# Increased circulating levels of Factor H-Related Protein 4 are strongly associated with age-related macular degeneration

Valentina Cipriani [1,2,3,4,16]*, Laura Lorés-Motta[5,16], Fan He [6], Dina Fathalla [7], Viranga Tilakaratna[6], Selina McHarg[6], Nadhim Bayatti[6], İlhan E. Acar [5], Carel B. Hoyng[5], Sascha Fauser[8,9], Anthony T. Moore[2,3,10], John R.W. Yates[2,3,11], Eiko K. de Jong [5], B. Paul Morgan[7,17], Anneke I. den Hollander [5,12,17], Paul N. Bishop[6,13,17] & Simon J. Clark [6,14,15,17]*

Age-related macular degeneration (AMD) is a leading cause of blindness. Genetic variants at the chromosome 1q31.3 encompassing the complement factor H (*CFH*, FH) and *CFH* related genes (*CFHR1-5*) are major determinants of AMD susceptibility, but their molecular consequences remain unclear. Here we demonstrate that FHR-4 plays a prominent role in AMD pathogenesis. We show that systemic FHR-4 levels are elevated in AMD ($P$-value = $7.1 \times 10^{-6}$), whereas no difference is seen for FH. Furthermore, FHR-4 accumulates in the choriocapillaris, Bruch's membrane and drusen, and can compete with FH/FHL-1 for C3b binding, preventing FI-mediated C3b cleavage. Critically, the protective allele of the strongest AMD-associated *CFH* locus variant rs10922109 has the highest association with reduced FHR-4 levels ($P$-value = $2.2 \times 10^{-56}$), independently of the AMD-protective *CFHR1–3* deletion, and even in those individuals that carry the high-risk allele of rs1061170 (Y402H). Our findings identify FHR-4 as a key molecular player contributing to complement dysregulation in AMD.

[1] William Harvey Research Institute, Clinical Pharmacology, Queen Mary University of London, London EC1M 6BQ, UK. [2] UCL Institute of Ophthalmology, University College London, London EC1V 9EL, UK. [3] Moorfields Eye Hospital NHS Foundation Trust, London EC1V 2PD, UK. [4] UCL Genetics Institute, University College London, London WC1E 6BT, UK. [5] Department of Ophthalmology, Donders Institute for Brain, Cognition and Behaviour, Radboud University Medical Center, Nijmegen 6525 GA, The Netherlands. [6] Division of Evolution and Genomic Sciences, Faculty of Biology Medicine and Health, School of Biological Sciences, University of Manchester, Oxford Road, Manchester M13 9PT, UK. [7] Systems Immunity URI, Division of Infection and Immunity, and UK DRI Cardiff, School of Medicine, Cardiff University, Cardiff CF14 4XN, UK. [8] Department of Ophthalmology, University Hospital of Cologne, Cologne 50924, Germany. [9] Roche Pharma Research and Early Development, F. Hoffmann-La Roche Ltd, Basel 4070, Switzerland. [10] Ophthalmology Department, University of California San Francisco, San Francisco, CA, USA. [11] Department of Medical Genetics, University of Cambridge, Cambridge CB2 0QQ, UK. [12] Department of Human Genetics, Donders Institute for Brain, Cognition and Behaviour, Radboud University Medical Centre, Nijmegen 6525 HR, The Netherlands. [13] Manchester Royal Eye Hospital, Manchester University NHS Foundation Trust, Manchester Academic Health Science Centre, Manchester M13 9WL, UK. [14] The Lydia Becker Institute of Immunology and Inflammation, Faculty of Biology, Medicine and Health, University of Manchester, Manchester, UK. [15] Present address: Department of Ophthalmology, Research Institute of Ophthalmology, Eberhard Karls University of Tübingen, 72076 Tübingen, Germany. [16] These authors contributed equally: Valentina Cipriani, Laura Lorés-Motta. [17] These authors jointly supervised this work: B. Paul Morgan, Anneke I. den Hollander, Paul N. Bishop, Simon J. Clark. *email: v.cipriani@qmul.ac.uk; simon.clark@uni-tuebingen.de

Age-related macular degeneration (AMD) is the most common cause of vision loss in Western societies[1]. Soft drusen are an early sign of AMD. These deposits form within Bruch's membrane (BrM) underneath the retinal pigment epithelium (RPE) basement membrane and contain apolipoprotein B and E, cholesterol-rich lipoproteins that are thought to be derived from the RPE[2]. In addition, they contain a variety of other proteins, with complement proteins being a prominent component[3]. This early stage of disease can then progress to late AMD, manifesting as either geographic atrophy ('dry' AMD) or choroidal neovascularization ('wet' AMD)[4].

AMD has a strong genetic basis; associations with 45 common single-nucleotide polymorphisms (SNPs) and 7 rare variants across 34 genetic loci have been reported in the largest genome-wide association study (GWAS) to date, explaining ~34% of AMD risk[5]. Many of these variants reside in genes encoding complement system components, particularly those encoded at the regulators of complement activation (RCA) locus on chromosome 1q31.3, including factor H (FH; *CFH*) and FH-related 1–5 (*CFHR1–5*)[6,7]. Common SNPs within *CFH*, including rs1061170 encoding a tyrosine to histidine substitution at position 402 (Y402H), were first identified as major susceptibility variants for AMD[8–11]. The recent largest GWAS established eight independent signals (four common variants, four rare) over 578 Mb of the RCA locus[5]. Except for the highly penetrant *CFH* missense variant R1210C[12] and synonymous variant rs35392876 in *CFH*, all variants are non-coding: four intronic in *CFH* (2), *CFHR5* (1) and *KCNT2* (1) and two intergenic (8 kb upstream *CFH*/35 kb downstream *KCNT2*; 14 kb downstream *CFHR1*/156 kb upstream *CFHR4*). The role of these genes in the pathogenesis of AMD is unclear.

The *CFH* gene encodes FH and its smaller splice variant, FH-like 1 (FHL-1)[13,14]. FH is the main plasma complement regulator, but FHL-1 predominates in BrM and choriocapillaris[6,15]. While FH/FHL-1 downregulate complement activation in plasma and on surfaces, the FHR proteins can compete with FH/FHL-1 for surface and ligand binding, thus disrupting their negative regulatory function and facilitate local activation[16,17] (see Fig. 3 of ref. [17] for an explanatory diagram of *CFH* and *CFHR* genes and the structures of FH, FHL-1 and FHR proteins). However, due to the extremely high level of sequence homology shared by all of the FHR proteins[17], it has thus far remained difficult to investigate their individual tissue expression patterns. Rare AMD-associated coding variants in *CFH* and their functional consequences directly implicate FH in the pathogenesis of AMD[5,12,18–22]. The molecular basis of the association of FH/FHL-1 402H variant to AMD pathology has been reported to involve altered binding to heparan sulfate, C-reactive protein or malondialdehyde, impacting local complement activation and subretinal inflammation[23–26]. Downstream of *CFH*, a common ~84 kb deletion of *CFHR3* and *CFHR1* and a rare ~120 kb deletion encompassing *CFHR1* and *CFHR4* are associated with reduced AMD risk, supporting the hypothesis that multiple genes at the locus may be involved in AMD[27–34]. In line with the genetic findings, dysregulation of the complement system in the eye and blood has been reported in the early stages of AMD predominating in the extracellular matrix (ECM) surrounding the fenestrated capillaries of the choriocapillaris that underlies BrM[35–38].

A recent GWAS identified an intronic variant in *CFHR4* that associated with increased systemic complement activation and AMD risk[39]. Furthermore, it has recently been reported that the top AMD-associated *CFH* variant rs10922109[5] is associated with altered *CFHR4* expression in the liver[40]. Taken together, these studies propose that, as well as FH, FHR-4 may also be involved in AMD. Having recently generated a novel, specific monoclonal antibody (mAb) against FHR-4, here we investigate, using a combination of biochemical, immunohistochemical and genetic approaches, whether FHR-4 directly impacts AMD pathogenesis. We show, in two large, independent cohorts, that blood FHR-4 levels are elevated in AMD patients compared to controls. FHR-4 is present in areas of pathology in AMD retina, co-localizing with complement activation products. In vitro functional analyses show that FHR-4 binds C3 fragments and competes out the binding of the regulatory proteins FH and FHL-1. Genetic association analyses show that several of the established AMD risk variants at the *CFH* locus are associated with FHR-4 levels in the blood, a finding strongly supported by haplotype association analyses. Taken together, our findings implicate FHR-4 as a key driver of complement dysregulation in the AMD retina.

## Results

### Systemic FHR-4 levels are elevated in advanced AMD cases.
Systemic FHR-4 concentrations were measured in plasma and serum samples of 484 late AMD patients (geographic atrophy and/or choroidal neovascularization) and 522 phenotyped controls, collected within two independent AMD studies (Cambridge and European Genetic Database (EUGENDA); Table 1). AMD patients had significantly elevated FHR-4 levels compared to controls, in each study separately ($\beta = 0.18$ and $P$ value $= 0.016$ for Cambridge and $\beta = 0.19$ and $P$ value $= 1.7 \times 10^{-4}$ for EUGENDA; Wald test) and in the two-cohort meta-analysis ($\beta = 0.19$, 95% confidence interval (CI) 0.11–0.27 and $P$ value $= 7.1 \times 10^{-6}$) (Table 1 and Fig. 1a). Association of FHR-4 levels stratified by type of end-stage disease, that is, choroidal neovascularization (CNV) only and geographic atrophy (GA) only, was additionally performed. These analysis showed comparable estimates in both cohorts (CNV only: $\beta = 0.15$ and $P$ value $= 0.068$ for Cambridge and $\beta = 0.18$ and $P$ value $= 0.001$ for EUGENDA; GA only: $\beta = 0.20$ and $P$ value $= 0.099$ for Cambridge and $\beta = 0.45$ and $P$ value $= 0.008$ for EUGENDA; Wald test) and in the meta-analysis (CNV only: $\beta = 0.17$, CI 0.09–0.26 and $P$ value $= 9.3 \times 10^{-5}$; GA only: $\beta = 0.28$, CI 0.09–0.47 and $P$ value $= 0.004$), with wider CIs for the GA-only group reflecting the smaller sample size (62 GA-only cases in Cambridge and 10 GA-only cases in EUGENDA). The overall adjusted odds ratio (OR) of advanced disease for an FHR-4 increase of 1 standard deviation was 1.37 (CI $= 1.19$–1.58; $P$ value $= 1.8 \times 10^{-5}$) (Supplementary Fig. 1a). We also measured systemic FH levels and found no significant difference between patients and controls ($P$ values 0.959, 0.535 and 0.704 for Cambridge, EUGENDA and meta-analysis, respectively; Table 1, Fig. 1b and Supplementary Fig. 1b).

### CFHR4 is expressed in the liver but not in the eye.
We found no evidence of transcription of the *CFHR4* gene in primary human RPE cells by reverse transcription PCR (Supplementary Fig. 2a). Analysis of the Gene Expression Omnibus datasets (https://www.ncbi.nlm.nih.gov/geo) confirmed absence of *CFHR4* transcription in the neurosensory retina, RPE and choroid using Affymetrix U133plus2 human genome arrays[41], Affymetrix Human Exon 1.0 ST arrays[42,43] or RNA sequencing[44,45] (Supplementary Fig. 2b–f). Analysis of gene expression across 53 human tissues from the Genotype-Tissue Expression project (https://www.ebi.ac.uk/gxa/home)[46] demonstrated that *CFHR4* expression was restricted to the liver (Supplementary Fig. 2g).

### FHR-4 in the choriocapillaris is associated with complement activation.
Immunostaining demonstrated that FHR-4 accumulates in the intercapillary septa, the extracellular matrix (ECM) between the fenestrated capillaries of the choriocapillaris (Fig. 2a–c) and within BrM (Fig. 2c). Diffusion experiments

**Table 1 Demographics of study cohorts and association analyses between AMD and systemic FHR-4/FH levels.**

| | Cambridge Controls | Cambridge Cases | EUGENDA Controls | EUGENDA Cases | Cambridge association with AMD, β, SE, p[b] | EUGENDA association with AMD, β, SE, p[b] | Meta-analysis β, 95% CI, p[b] |
|---|---|---|---|---|---|---|---|
| N | 214 | 304 | 308 | 180 | | | |
| Age (years) (SD) | 75.2 (8.0) | 74.1 (8.3) | 70.0 (6.5) | 79.3 (8.6) | | | |
| Male (%) | 36.5 | 47.0 | 42.9 | 42.2 | | | |
| AMD phenotype | | | | | | | |
| CNV only | | 191 | | 156 | | | |
| GA only | | 62 | | 10 | | | |
| Mixed | | 51 | | 14 | | | |
| FHR-4 levels, μg ml$^{-1}$ (95% CI)[a] | 5.5 (4.9–6.2) | 6.6 (6.0–7.2) | 6.0 (5.6–6.3) | 7.2 (6.6–7.8) | 0.18, 0.07, 0.016 (0.17, 0.07, 0.018) | 0.19, 0.05, $1.7 \times 10^{-4}$ (0.24, 0.06, $8.4 \times 10^{-5}$) | 0.19, 0.11–0.27, $7.1 \times 10^{-6}$ (0.21, 0.12–0.30, $4.8 \times 10^{-6}$) |
| FH levels, μg ml$^{-1}$ (95% CI)[a] | 349.0 (338.9–359.4) | 348.6 (340.2–357.2) | 304.7 (297.3–312.2) | 308.7 (298.0–319.8) | −0.001, 0.2, 0.959 (0.006, 0.02, 0.752) | 0.01, 0.02, 0.535 (0.02, 0.02, 0.433) | 0.01, −0.02 to 0.03, 0.704 (0.01, −0.02 to 0.04, 0.466) |

[a]FHR-4 and FH levels are expressed as geometric mean values (back-log transformed).
[b]Wald tests using linear regression models; adjusted estimates for sex, age, batch effects and first two genetic principal components are displayed within parentheses.

demonstrated that FHR-4 does not completely transit this ECM (Supplementary Fig. 3). Drusen, a hallmark of AMD, were strongly positive for FHR-4 Ab labelling (Fig. 2d). C3b also localized to the choriocapillaris intercapillary septa and appeared to co-localize with FHR-4 (Fig. 2e). FHR-4 is reported to bind C3b and stabilize the C3 convertase[47,48]. We confirmed that FHR-4 binds immobilized C3b (Fig. 2f) and demonstrated that FHR-4 competes with the negative regulators, FH and FHL-1, for binding immobilized C3b (Fig. 2g). The consequences of this were modelled in vitro employing C3b α-chain cleavage assays (Fig. 2h and Supplementary Fig. 4). C3b was incubated with FHL-1 and factor I (FI) titrated to give ~80% C3b α-chain cleavage; FHR-4 inhibited α-chain cleavage in a dose-dependent manner; a 2.5-fold molar excess of FHR-4 over FHL-1 caused 50% reduction in cleavage (Fig. 2i).

**CFH locus AMD risk variants associate with FHR-4 levels.** The International AMD Genomics Consortium (IAMDGC) GWAS[5] reported eight independently associated variants at the CFH locus (Fig. 3a and Supplementary Data 1). We repeated single-variant association analyses with AMD in the Cambridge and EUGENDA samples (originally part of the IAMDGC dataset) and observed all ORs with the same direction and similar magnitude as in IAMDGC at all variants, except for rare variant rs191281603 (Supplementary Data 1).

We hypothesized that one or several of the established AMD risk variants at the CFH locus are associated with increased systemic FHR-4 levels. The rare CFH variant R1210C[12], present heterozygously in a single case from the Cambridge cohort (with corresponding values of FHR-4 and FH levels equal to 5.7 and 296.4, respectively), was excluded from this analysis. The top (rs10922109, 1.1), second (rs570618, 1.2; proxy for Y402H), fifth (rs187328863, 1.5) and sixth (rs61818925, 1.6) IAMDGC hits at the CFH locus showed strong associations with FHR-4 levels (after Bonferroni correction for multiple testing), with direction of allelic effect on levels concordant with that on disease for all variants (Table 2, Fig. 3b, Supplementary Data 2 and Supplementary Fig. 5). The strongest allelic effect on FHR-4 levels was seen at the top IAMDGC variant rs10922109, with $\beta = -0.42$ and P value $= 2.2 \times 10^{-56}$ (Wald test) for the minor allele A associated with decreased disease risk. In the Cambridge and EUGENDA cohorts, respectively, this finding translates into (back-log transformed) FHR-4 levels expressed as geometric mean values [95% CIs] equal to 7.7 μg ml$^{-1}$ [7.0–8.5] and 8.5 μg ml$^{-1}$ [7.9–9.1] in CC genotype individuals, 5.5 μg ml$^{-1}$ [5.0–6.1] and 6.0 μg ml$^{-1}$ [5.7–6.4] in AC genotype individuals and 3.2 μg ml$^{-1}$ [2.5–4.0] and 3.6 μg ml$^{-1}$ [3.3–3.9] in AA genotype individuals. Analogous single-variant association analyses with FH levels revealed a significant association only at rs10922109 and rs61818925 with much smaller effect size ($\beta = 0.03$ and $-0.03$, respectively) (Table 2, Supplementary Data 2 and Supplementary Fig. 5).

To assess whether genetic variants at loci other than CFH associated with systemic FHR-4 levels, we performed a subsidiary (hypothesis-free) GWAS meta-analysis of FHR-4 levels. A single ~1 Mb region spanning the extended CFH locus (chr1q31.3:196,240,335–197,281,307) showed genome-wide significant ($P \leq 5 \times 10^{-8}$) associations with FHR-4 levels (Fig. 4a, Supplementary Fig. 6a and Supplementary Data 3 and Supplementary Data 4). The top signal rs7535263 is in tight linkage disequilibrium (LD) ($R^2 = 0.98$, $D' = 1.00$) with the top IAMDGC variant rs10922109 (1.1) (regional plot in Fig. 4a, Supplementary Data 5; OR = 0.11, P value $= 1.7 \times 10^{-612}$ in IAMDGC[5]). Analogous GWAS meta-analysis of FH levels also revealed a single genome-wide significant association confined to

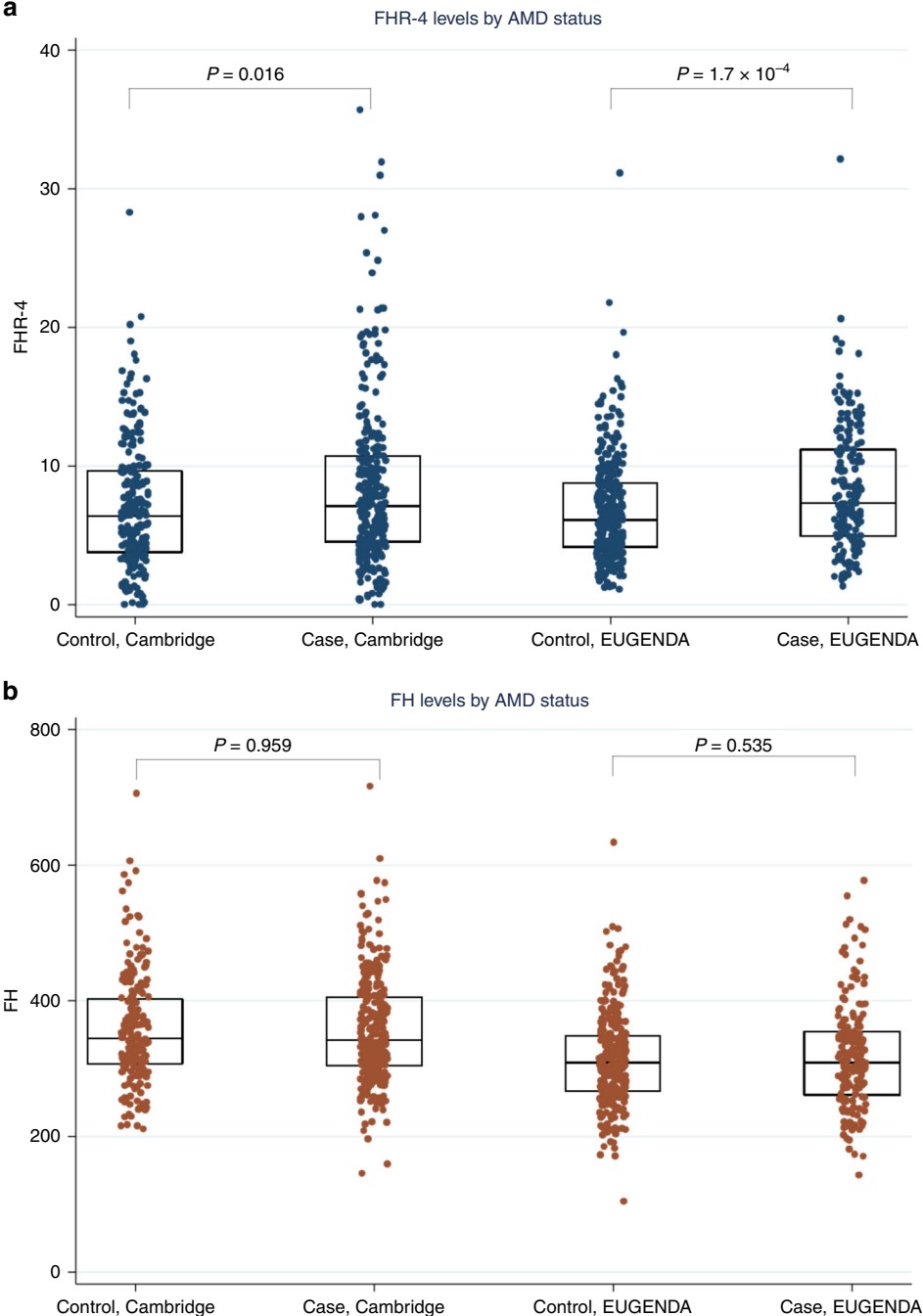

**Fig. 1 Systemic FHR-4 levels are elevated in AMD patients. a** shows box plots (and corresponding data points) of FHR-4 levels measured in two separate AMD cohorts: Cambridge (plasma from 214 controls and 304 late AMD cases) and EUGENDA (serum from 308 controls and 180 late AMD cases). AMD patients show statistically significantly elevated FHR-4 levels compared to controls. Geometric mean FHR-4 levels were: Cambridge, 5.5 µg ml$^{-1}$ in controls vs. 6.6 µg ml$^{-1}$ in cases; EUGENDA, 6.0 µg ml$^{-1}$ in controls vs. 7.2 µg ml$^{-1}$ in cases. These differences remained significant after adjustment for sex, age, batch effects and first two genetic principal components ($P$ value = 0.018 and 8.4 × 10$^{-5}$ for Cambridge and EUGENDA, respectively; Wald test). **b** shows box plots (and corresponding data points) of FH levels measured in the same samples, where no statistically significant difference between cases and controls was observed: Cambridge, 349.0 µg ml$^{-1}$ in controls vs. 348.6 µg ml$^{-1}$ in cases; EUGENDA, 304.7 µg ml$^{-1}$ in controls vs. 308.7 µg ml$^{-1}$ in cases. Each box plot depicts median value (central line), first quartile (lower bound line) and third quartile (upper bound line). Source data are provided as a Source Data file.

variants in tight LD in a ~150 kb region at the *CFH* locus (chr1q31.3:196,674,714–196,825,287; including rs6677604, a proxy for the previously reported AMD protective *CFHR1–3* deletion[29]) (Fig. 4b, Supplementary Fig. 6b, Supplementary Data 6, Supplementary Data 7 and Supplementary Data 8), but effect on FH levels was limited ($\beta = -0.10$, $P$ value = 2.4 × 10$^{-11}$,

Wald test, at the top variant rs74696321). Notably, the intronic AMD risk variant rs6685931 in *CFHR4* (LD with rs10922109: $R^2 = 0.43$, $D' = 0.96$), associated with complement activation in the recent GWAS[39], was strongly associated with levels of FHR-4 ($\beta = 0.28$, $P$ value = 2.3 × 10$^{-25}$, Wald test), but not FH ($\beta = 0.005$, $P$ value = 0.607, Wald test).

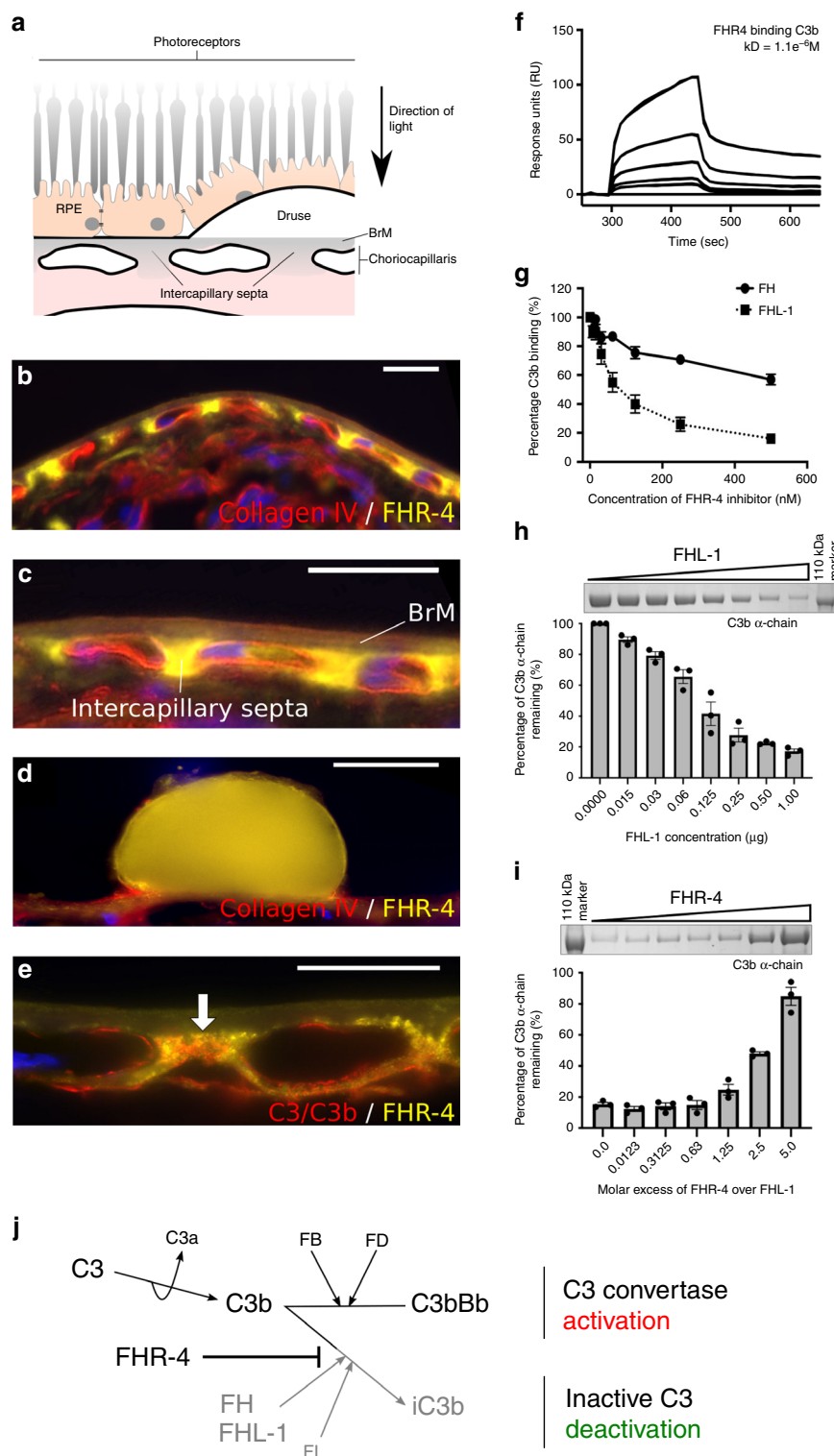

**CFH locus haplotypes strongly associate with AMD and FHR-4 levels**. To assess the combined effect of variants at the *CFH* locus, we carried out association analyses of the haplotypes formed by the seven *CFH* variants considered in our study with AMD and FHR-4/FH levels; we included rs6677604 as proxy for the *CFHR1–3* deletion[29] to assess its influence on FHR-4/FH levels. The rare *CFHR1-4* deletion[33,34] was present heterozygously in three controls and one advanced AMD patient and was not included in this analysis. Haplotype associations with AMD were also assessed in the whole IAMDGC dataset[5].

We observed nine common haplotypes with overall frequency ≥1% (Fig. 5b and Supplementary Data 9). The most frequent haplotype CTTGCCGC (H1; controls 32%, cases 49% in IAMDGC) that carries the disease risk allele of the proxy for Y402H (1.2) was used as reference. Common H2–H5 and rarer H7 haplotypes carried significantly lower AMD risk than H1, while rarer H6 (TTTGCCGC) and H9 (CTTGCTGC) carried higher risk than H1; H8 (CTTGCCTC) did not show a significantly different risk from H1 (Fig. 5a and Supplementary Data 9). Similar OR estimates were observed

**Fig. 2 Accumulation of FHR-4 in the choriocapillaris inhibits C3b breakdown. a** shows a schematic diagram illustrating anatomical structures in the macula including the retinal pigment epithelium (RPE), the underlying Bruch's membrane (BrM) and the intercapillary septa within the choriocapillaris; basement membranes are represented as black lines. Drusen, hallmark lesions of early AMD, form within BrM underneath the RPE basement membrane. **b, c** Immunohistochemistry showing the localization of FHR-4 (yellow) predominantly in the intercapillary septa: weak labelling is also seen within BrM. Collagen IV staining is used to delineate basement membranes, which define the inner and outer borders of BrM (red); DAPI labelling is in blue. FHR-4 is also localized in drusen (**d**); the RPE is absent from these tissue sections. **e** Both FHR-4 and C3/C3b localize in the intercapillary septa of the choriocapillaris (white arrow): scale bars 20 μm. SPR analysis showing the binding of FHR-4 to immobilized C3b (**f**). Solid-phase binding assays demonstrate that FHR-4 can compete off fluid-phase FH or FHL-1 binding to immobilized C3b (**g**). Measurement of FHL-1-mediated breakdown of C3b by factor I (**h**); in the presence of fixed concentrations of C3b and factor I, increasing concentrations of FHL-1 result in increased breakdown of the C3b α-chain (see Supplementary Fig. 4 for full gel image). **i** optimal C3b breakdown conditions from **h** are repeated, but now include increased concentrations of fluid-phase FHR-4, where an inhibition of FHL-1/FI-mediated C3b α-chain breakdown is observed (see Supplementary Fig. 4 for full gel image). **j** FHR-4 prevents FHL-1 acting as a cofactor for factor I, this results in the formation of a C3 convertase and the activation of the amplification loop of complement and subsequent inflammation. Data in **g–i** are from $n = 3$, averaged from three independent experiments ± s.e.m. Source data are provided as a Source Data file.

in our two-cohort meta-analysis (Fig. 5a and Supplementary Data 9).

Haplotypes H2 (CTGGACTC) and H3 (CTGAACGC) strongly associated with decreased FHR-4 levels and carry independent effects with no overlapping CIs ($\beta = -0.49$, P value = $1.7 \times 10^{-44}$ and $\beta = -0.25$, P value = $4.4 \times 10^{-10}$, respectively; Wald test) (Fig. 5a and Supplementary Data 9). While both haplotypes carry the FHR-4-lowering/AMD-protective alleles A of rs10922109 (1.1) and G of rs570618 (1.2), H2 carries the FHR-4-lowering/AMD-protective allele T of rs61818925 (1.6) and H3 carries the FHR-4-lowering/AMD-protective allele A of rs6677604, tag for the AMD-protective *CFHR1–3* deletion. Neither of the haplotypes showed a more significant association with FHR-4 levels than the meta-analysis single-variant associations (Table 2). Analogous haplotype association analyses with FH levels revealed a significant association only at H2 (after Bonferroni correction) with small effect ($\beta = 0.07$, P value = $3.3 \times 10^{-6}$, Wald test). Results for the diplotype (haplotype pair) association analyses are shown in Supplementary Data 10 and Supplementary Fig. 7. Remarkably, among the genotypes that contain one copy of H1 (Y402H), diplotypes H1:H2, H1:H3 and H1:H7 showed a significantly lower AMD risk (OR = 0.33, P value = $5.3 \times 10^{-152}$, OR = 0.29, P value = $1.0 \times 10^{-161}$ and OR = 0.42, P value = $2.2 \times 10^{-24}$, respectively, in IAMDGC; Wald test) and decreased levels of FHR-4 ($\beta = -0.54$, P value = $2.0 \times 10^{-16}$, $\beta = -0.31$, P value = $8.0 \times 10^{-6}$ and $\beta = -0.54$, P value = 0.001, respectively, in our two-cohort meta-analysis), compared to reference H1:H1 genotype.

Using a sequential forward approach, we tested the association of the haplotypes formed by rs10922109 (1.1) and rs61818925 (1.6), the best two single-variant association signals with FHR-4 levels in our meta-analysis (Table 2). The most frequent haplotype CG (H1*; controls 44%, cases 64% in IAMDGC) was used as reference. We observed three other haplotypes (H2*–H4*) carrying both distinct AMD lower risk (in IAMDGC; with similar OR estimates in our two-cohort meta-analysis) and distinct lowering effects on FHR-4 levels (Fig. 6a–c and Supplementary Data 11). Haplotype H2* (AT) showed the strongest association with FHR-4 levels ($\beta = -0.52$, P value = $2.4 \times 10^{-58}$, Wald test) with a larger effect size and more significant P value than any of the single-variant signals (Table 2). Haplotype H4* (AG) was the only haplotype also associated with FH levels ($\beta = 0.08$, P value = $7.7 \times 10^{-7}$, Wald test). Adding SNP rs570618 (1.2), the third meta-analysis single-variant association signal with FHR-4 levels (Table 2), to the inferred haplotypes did not significantly improve the dissection of the genetic effects on FHR-4 levels at the *CFH* locus (lowest P value = $2.0 \times 10^{-53}$, Wald test, at haplotype GAT with $\beta = -0.50$).

## Discussion

Here we provide compelling evidence to show that AMD is associated with genetically driven elevated circulating levels of FHR-4 and not associated with circulating FH levels. FHR-4 likely predisposes to disease by penetrating the ECM of the choriocapillaris and BrM and acting locally by facilitating complement activation. FHL-1 is the complement regulator primarily responsible for protecting intercapillary septa ECM from complement activation[6,15], but this protective function may be inhibited by FHR-4. FHR-4 accumulates in the intercapillary septa of the choriocapillaris, the ECM surrounding the fenestrated capillaries and a major site of AMD pathogenesis (Fig. 2a–e). *CFHR4* gene transcription was absent in the RPE and choroid, demonstrating that the systemic circulation is the source of FHR-4 in the eye. Deposition of C3b in the intercapillary septa will result in C3 convertase formation, complement activation and inflammation unless sufficiently regulated by FI-mediated C3b breakdown in the presence of FHL-1[15]. Based on our in vitro competition assays (Fig. 2j), we propose that in AMD the accumulation of FHR-4 in the ECM out-competes FHL-1 for C3b binding, thereby preventing FI-mediated C3b breakdown and driving complement activation. FHR-4 bound to deposited C3b may also directly facilitate C3 convertase formation[47,48]. Excessive complement turnover, driven by FHR-4 accumulation, will continue to recruit and activate circulating immune cells[49], another key feature of early AMD. Quite how complement over-activation leads to drusen formation remains unclear, although studies have demonstrated that a combination of both complement over-activation and oxidative stress can result in lipid accumulation in RPE cells and BrM[50]. Furthermore, non-canonical roles of complement have also been shown to influence the ability to clear apolipoproteins from RPE cells and BrMs in various animal models[51].

Remarkably, the *CFH* locus was the only genome-wide significant locus in our GWAS meta-analysis of FHR-4 levels. The top signal is in tight LD with the strongest published AMD association signal at the *CFH* locus[5] (Fig. 4a, Supplementary Fig. 6a and Supplementary Data 3, Supplementary Data 4 and Supplementary Data 5). The triangular relationship between established susceptibility *CFH* locus variants, FHR-4 levels and AMD provides strong support for the association we observe between FHR-4 levels and increased AMD risk (Table 1, Fig. 1a and Supplementary Fig. 1a) to be causal. Our haplotype-based association analyses allowed the individual effects of FHR-4 levels, the *CFHR1-3* deletion and the Y402H variant of FH/FHL-1 to be dissected. Using the most frequent haplotype H1 (carrying the risk allele of Y402H) as reference, the two most protective haplotypes, H2 and H3, were associated with the lowest levels of FHR-4 (Fig. 5a–b and Supplementary Data 9). The H2 haplotype

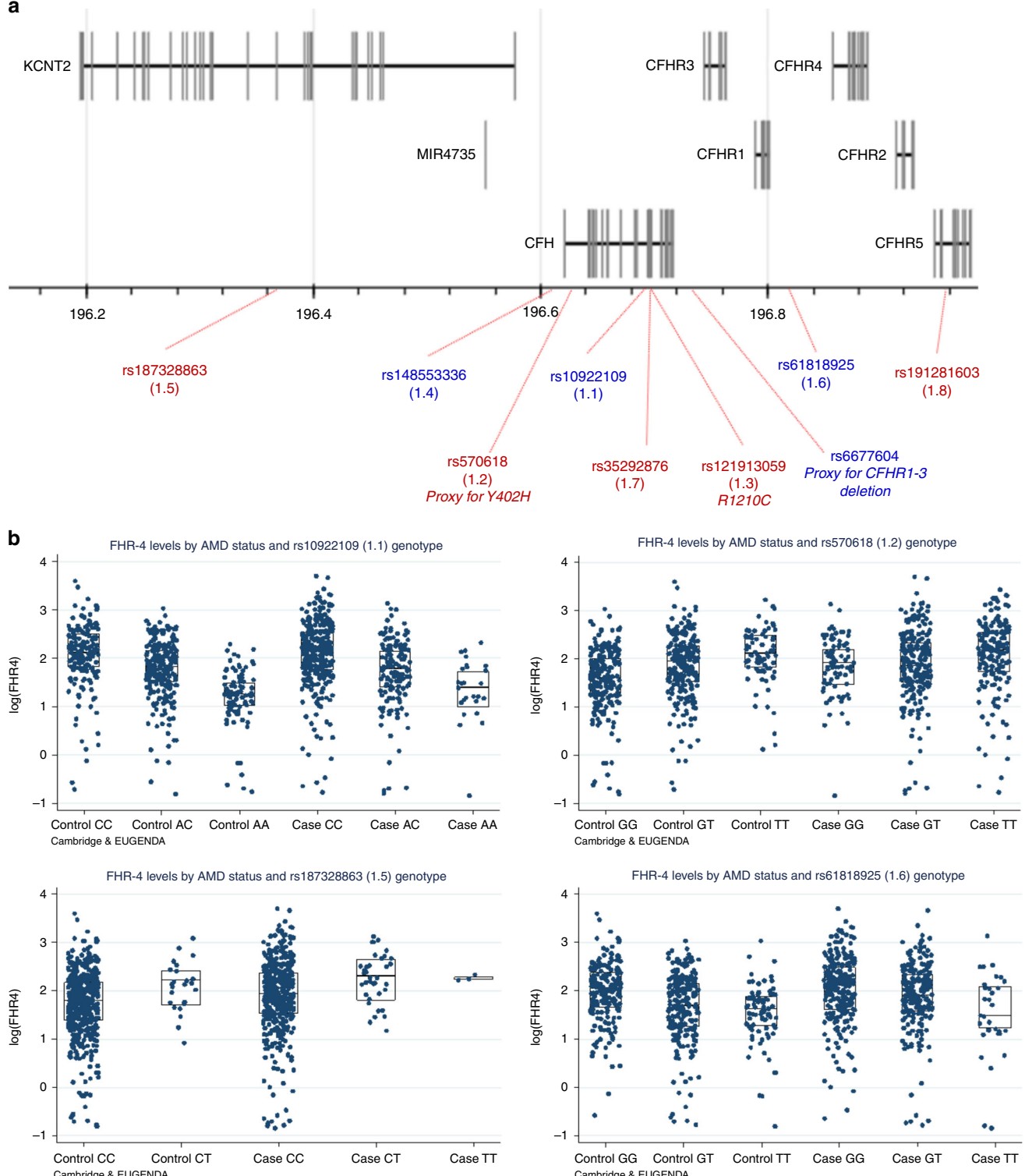

**Fig. 3 Four AMD risk variants at the *CFH* locus are strongly associated with FHR-4 levels.** Schematic diagram of chromosome 1 showing the genes in the *CFH* locus and the genomic location of the eight established AMD risk variants from the large IAMDGC GWAS of AMD[5] and rs6677604, a proxy for the previously reported AMD-protective *CFHR1–3* deletion[29] (**a**). Variant annotations are in red or blue depending on whether the corresponding minor allele is AMD deleterious or protective. The rare missense variant rs121913059 (1.3; R1210C) was only present heterozygously in a case individual from the Cambridge cohort, and therefore was not included in the genetic association analyses with the FHR-4/FH levels. **b** shows box plots (and corresponding data points) of FHR-4 levels by AMD status and SNP genotype for the four variants that showed significant associations (after Bonferroni correction) with FHR-4 levels (Table 2), in the Cambridge and EUGENDA cohorts combined. Each box plot depicts median value (central line), first quartile (lower bound line) and third quartile (upper bound line). Source data are provided as a Source Data file.

**Table 2 Single-variant association analyses with FHR-4 and FH levels for the eight AMD independently associated variants at the CFH locus from the IAMDGC study[5].**

| IAMDGC association signal number (direction[a]) | dbSNP ID Chr position[b] Major/minor allele (imputation R2)[c] | Cambridge MAF controls | Cambridge MAF cases | Cambridge Association with FHR-4 levels[d] β (SE) P | Cambridge Association with FH levels[d] β (SE) P | EUGENDA MAF controls | EUGENDA MAF cases | EUGENDA Association with FHR-4 levels[d] β (SE) P | EUGENDA Association with FH levels[d] β (SE) P | Meta-analysis Association with FHR-4 levels β (SE) P | Meta-analysis Association with FH levels β (SE) P |
|---|---|---|---|---|---|---|---|---|---|---|---|
| 1.1 (−) | rs10922109 1:196,704,632 C/A (1.00) | 0.415 | 0.208 | −0.43 (0.05) $5.8 \times 10^{-16}$ | 0.04 (0.01) 0.003 | 0.437 | 0.219 | −0.42 (0.03) $3.3 \times 10^{-35}$ | 0.02 (0.02) 0.318 | −0.42 (0.03) $2.2 \times 10^{-56}$ | 0.03 (0.01) 0.005 |
| 1.2 (+) | rs570618 1:196,657,064 G/T (1.00) | 0.367 | 0.599 | 0.20 (0.05) $3.8 \times 10^{-5}$ | −0.004 (0.01) 0.783 | 0.354 | 0.572 | 0.24 (0.03) $3.0 \times 10^{-12}$ | 0.01 (0.01) 0.669 | 0.23 (0.03) $1.6 \times 10^{-16}$ | 0.0008 (0.01) 0.933 |
| 1.3 (+) | rs121913059 1:196,716,375 C/T (genotyped) | No T allele carriers | Only 1 case heterozygote carrier | Not analysed | Not analysed | No T allele carriers | No T allele carriers | Not analysed | Not analysed | Not meta-analysed | Not meta-analysed |
| 1.4 (−) | rs148553336 1:196,613,173 T/C (genotyped) | 0.020 | 0.002 | 0.28 (0.27) 0.287 | −0.17 (0.07) 0.019 | 0.004 | No C allele carriers | Not analysed | Not analysed | Not meta-analysed | Not meta-analysed |
| 1.5 (+) | rs187328863 1:196,380,158 C/T (0.83) | 0.010 | 0.047 | 0.31 (0.15) 0.038 | −0.07 (0.04) 0.107 | 0.038 | 0.040 | 0.35 (0.10) $2.9 \times 10^{-4}$ | −0.07 (0.04) 0.089 | 0.34 (0.08) $2.8 \times 10^{-5}$ | −0.07 (0.03) 0.019 |
| 1.6 (−) | rs61818925 1:196,815,450 G/T (0.87) | 0.399 | 0.276 | −0.29 (0.06) $1.8 \times 10^{-7}$ | −0.01 (0.02) 0.642 | 0.393 | 0.315 | −0.29 (0.04) $3.3 \times 10^{-15}$ | −0.06 (0.02) $4.3 \times 10^{-4}$ | −0.29 (0.03) $2.8 \times 10^{-22}$ | −0.03 (0.01) 0.005 |
| 1.7 (+) | rs35292876 1:196,706,642 C/T (genotyped) | 0.005 | 0.016 | −0.05 (0.23) 0.815 | −0.11 (0.06) 0.090 | 0.008 | 0.025 | 0.32 (0.14) 0.019 | 0.04 (0.06) 0.517 | 0.22 (0.12) 0.057 | −0.03 (0.04) 0.500 |
| 1.8 (+) | rs191281603 1:196,958,651 C/G (0.42) | 0.009 | 0.007 | 0.11 (0.46) 0.812 | −0.09 (0.13) 0.490 | 0.010 | 0.008 | 0.23 (0.25) 0.357 | 0.24 (0.11) 0.025 | 0.20 (0.22) 0.357 | 0.10 (0.08) 0.198 |

*MAF* minor allele frequency, *Chr* chromosome, *SE* standard error, *IAMDGC* International Age-related Macular Degeneration Genomics Consortium.
[a]Direction of association with AMD for the minor allele, as estimated in the IAMDGC study[5].
[b]Chromosomal position is given according to the NCBI RefSeq hg19 human genome reference assembly. Bonferroni correction for multiple testing of 8 variants = 0.00625 (0.05/8).
[c]Imputation quality metric $R^2$ as estimated in the IAMDGC study[5].
[d]Wald tests using linear regression models adjusted for AMD status, sex, age, batch effects and the first two genetic principal components (as estimated within the IAMDGC study[5]).

(carrying the FHR-4-lowering/AMD-protective alleles A of rs10922109 (1.1) and T of rs61818925 (1.6)) does not contain the CFHR1–3 deletion, suggesting that lower FHR-4 levels confer the disease-protective effect. Furthermore, the diplotype analysis demonstrates that the H1:H2 genotype is associated with disease-protection relative to H1:H1, suggesting a dominant decreased disease risk effect of lower FHR-4 levels even in the presence of the Y402H risk variant on the other allele (Supplementary Data 10 and Supplementary Fig. 7). Finally, we showed that the two independently AMD-associated variants rs10922109 (1.1) and rs61818925 (1.6) are a minimal set of variants that explain the genetic effect on FHR-4 levels at the CFH locus (Fig. 6a–c).

FH levels were not different between cases and controls in our two independent cohorts (Fig. 1b and Supplementary Fig. 1b). Previous studies have measured systemic levels of FH in AMD and reported inconsistent results[52–60]. The sample size of our analysis (484 cases and 522 controls) exceeds all previous investigations. Our GWAS meta-analysis of FH levels reveals a similar genetic structure to that previously reported[52], with the top signal in high LD with variants that tag the common CFHR1–3 deletion (Fig. 4b, Supplementary Fig. 6b, Supplementary Data 6, Supplementary Data 7 and Supplementary Data 8). The data also show that systemic FH and FHR-4 levels are dictated by a different genetic architecture (Supplementary Fig. 8). The top signal for FH levels, rs74696321 ($\beta = -0.10$, P value $= 2.4 \times 10^{-11}$), is only among the genome-wide significant association tail for FHR-4 levels (653th hit, P value $= 7.4 \times 10^{-9}$) with opposite direction of allelic effect ($\beta = 0.23$), while the top signal for FHR-4 levels, rs7535263 ($\beta = -0.42$, P value $= 9.0 \times 10^{-57}$), tagging the top AMD-associated variant rs10922109, does not pass the genome-wide significance threshold in the GWAS meta-analysis of FH levels ($\beta = 0.03$, P value $= 0.005$). It should be noted that the circulating levels of FHR-4 are clearly associated with AMD risk, but the molar ratios of FHR-4 and FH/FHL-1 in blood are not representative of the ratios of the accumulated proteins in the ECM of the choriocapillaris and BrM. This can be attributed to the relatively large hydrodynamic size of FH compared to FHR-4 and FHL-1; we have previously shown that there is more FHL-1 in the tissue than FH, and that FH, unlike FHL-1, cannot diffuse across BrM[15,61]. Furthermore, the absence of local FHR-4 expression in the eye emphasizes the relevance of systemic levels of this protein for its accumulation in the choriocapillaris, whereas FHL-1, and any FH that is present, may be derived locally or systemically.

Genetically driven variations in the levels and functions of alternative pathway complement proteins play a central role in AMD pathogenesis. Common and rare coding variants in CFH are important: the common Y402H variant and a majority of the rare variants in CFH identified to date (that generally result in a familial, early-onset condition) affect the function of both FHL-1 and FH, suggesting a particular role for FHL-1 in AMD pathogenesis[62,63]. However, there are rare variants affecting only FH, including the R1210C mutation, strongly associated with early-onset AMD, showing that full-length FH also has an important role[12]. In addition, mutations in CFI and common variants in C3 and CFB modify AMD risk[5,64]. Therefore, it can be concluded that a balance between the actions of proteins that inhibit the alternative pathway (FH/FHL-1, FI) and those that activate the alternative pathway (C3, FB) influence AMD risk. Here we provide compelling data suggesting another regulator of the alternative pathway, FHR-4, is likely to have an important role in regulating this balance and thereby modifying AMD risk. This research implies that targeting FHR-4 may represent a future therapeutic avenue to explore in the treatment of AMD. Our demonstration that high systemic FHR-4 levels are associated with AMD risk makes the case for a therapy that lowers systemic

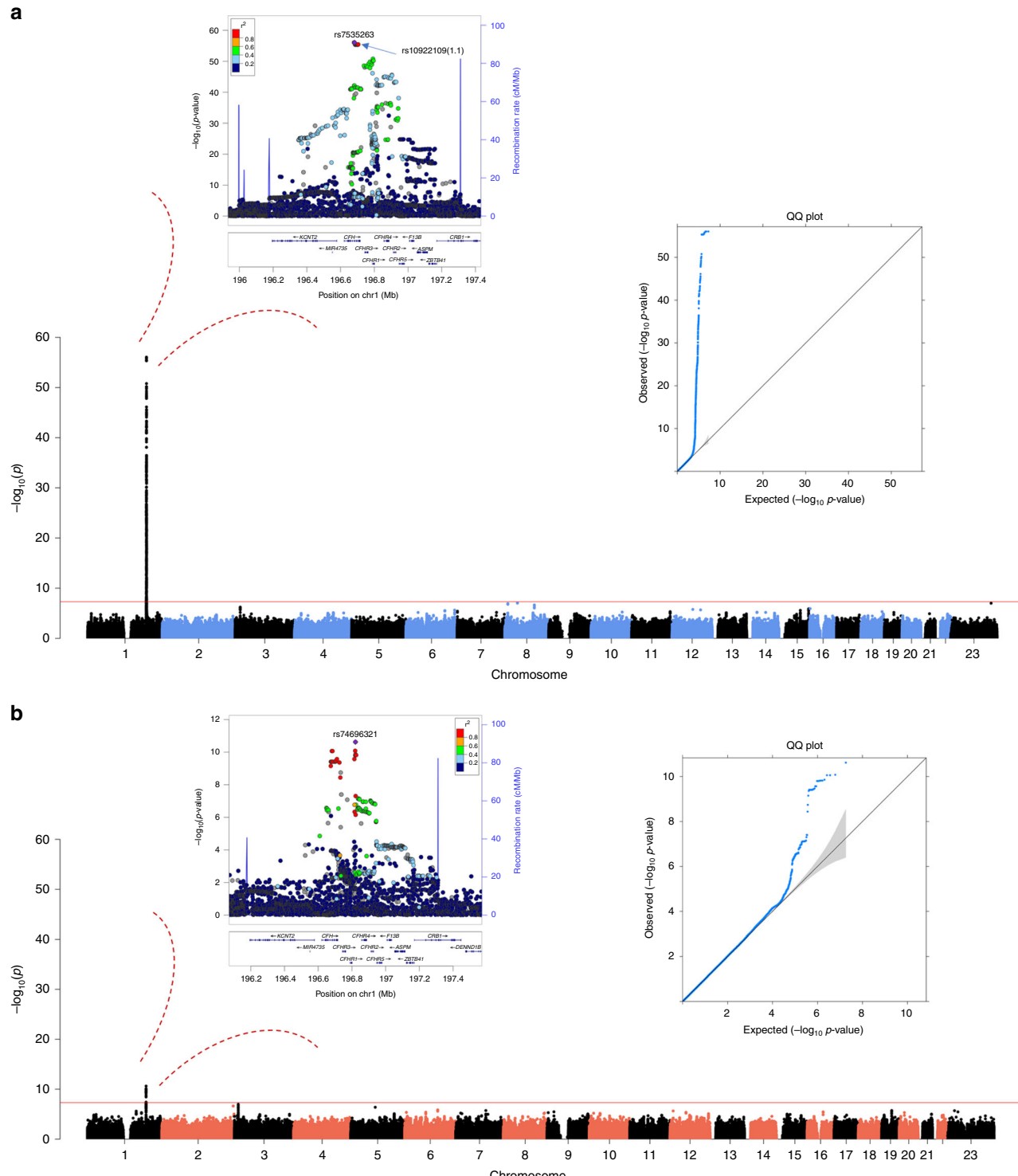

**Fig. 4 GWAS meta-analysis of FHR-4 levels reveals a strong signal spanning the *CFH* locus.** Each panel shows a Manhattan plot, a regional plot (upper left-hand side) and a quantile–quantile (QQ) plot (upper right-hand side) for the results of the GWAS meta-analysis of FHR-4 levels (**a**) and FH levels (**b**). Manhattan plots illustrate *P* values (Wald test) for each single variant tested for association with log(levels). Observed $-\log_{10}(P)$ values are plotted against the genomic position of each variant on chromosomes 1–22 plus the X chromosome. The horizontal red line indicates the threshold considered for genome-wide significance (*P* value $\leq 5 \times 10^{-8}$). Regional plots show the only genome-wide association signal observed, that is, at the *CFH* locus (on chromosome 1q31.3). The most associated variant is denoted by a purple circle and is labelled by its rsID. The other surrounding variants are shown by circles coloured to reflect the extent of LD with the most associated variant (based on 1000 Genomes data, November 2014). A diagram of the genes within the relevant regions is depicted below each plot. Physical positions are based on NCBI RefSeq hg19 human genome reference assembly. QQ plots compare the distribution of the observed test statistics with its expected distribution under the null hypothesis of no association. A marked departure from the null hypothesis (reference line) is seen in the meta-analysis of FHR4 levels. Genomic inflation values (λ) were equal to 1.008 and 1.005 from the GWASs of FHR-4 levels and 1.002 and 1.014 from the GWASs of FH levels, in the Cambridge and EUGENDA studies, respectively.

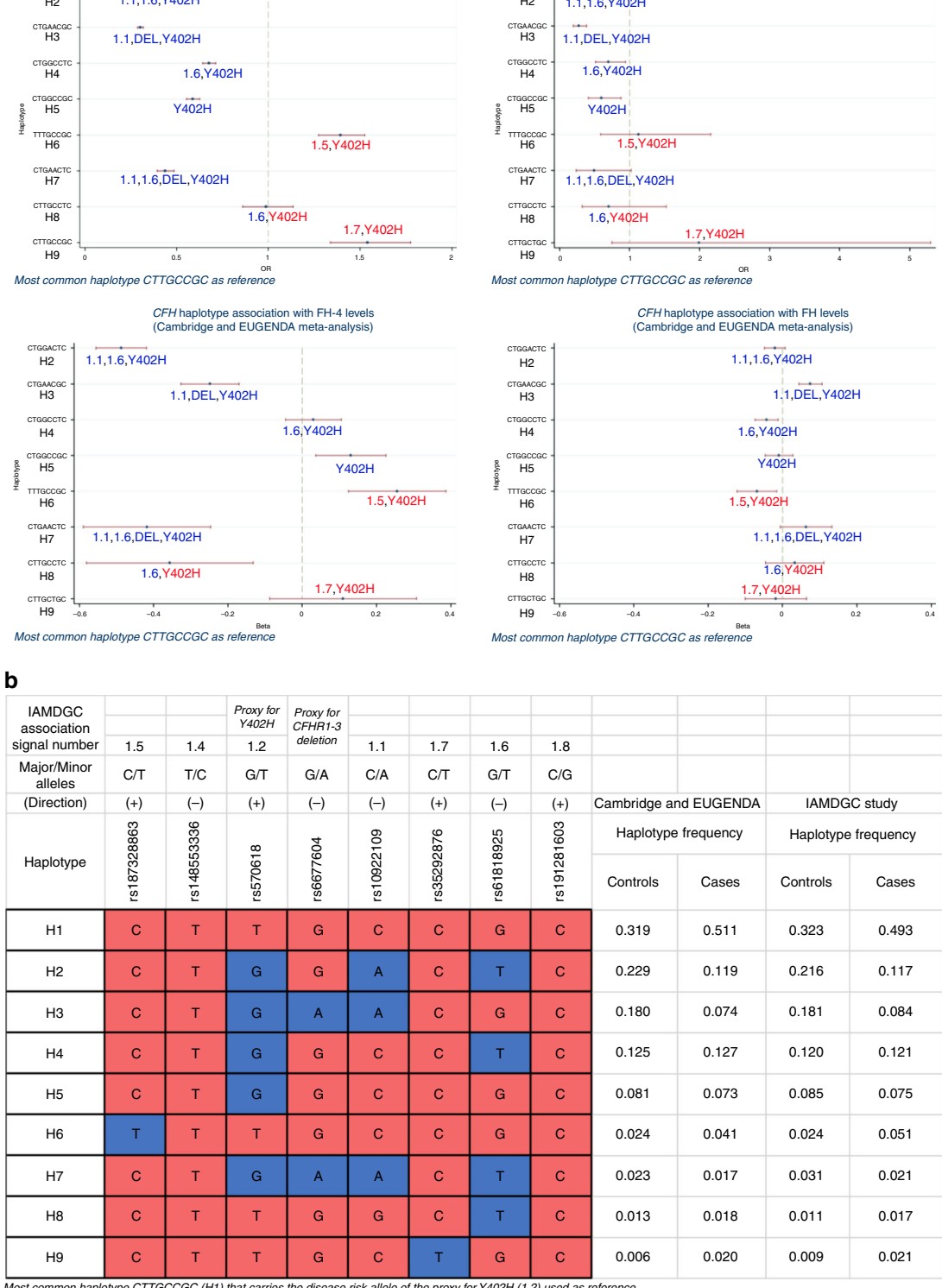

**a**

*CFH haplotype association with AMD (IAMDGC study)* — Most common haplotype CTTGCCGC as reference

*CFH haplotype association with AMD (Cambridge and EUGENDA meta-analysis)* — Most common haplotype CTTGCCGC as reference

*CFH haplotype association with FH-4 levels (Cambridge and EUGENDA meta-analysis)* — Most common haplotype CTTGCCGC as reference

*CFH haplotype association with FH levels (Cambridge and EUGENDA meta-analysis)* — Most common haplotype CTTGCCGC as reference

**b**

| IAMDGC association signal number | 1.5 | 1.4 | *Proxy for Y402H* 1.2 | *Proxy for CFHR1-3 deletion* | 1.1 | 1.7 | 1.6 | 1.8 | Cambridge and EUGENDA | | IAMDGC study | |
|---|---|---|---|---|---|---|---|---|---|---|---|---|
| Major/Minor alleles | C/T | T/C | G/T | G/A | C/A | C/T | G/T | C/G | Haplotype frequency | | Haplotype frequency | |
| (Direction) | (+) | (−) | (+) | (−) | (−) | (+) | (−) | (+) | | | | |
| Haplotype | rs187328863 | rs148553336 | rs570618 | rs6677604 | rs10922109 | rs35292876 | rs61818925 | rs191281603 | Controls | Cases | Controls | Cases |
| H1 | C | T | T | G | C | C | G | C | 0.319 | 0.511 | 0.323 | 0.493 |
| H2 | C | T | G | G | A | C | T | C | 0.229 | 0.119 | 0.216 | 0.117 |
| H3 | C | T | G | A | A | C | G | C | 0.180 | 0.074 | 0.181 | 0.084 |
| H4 | C | T | G | G | C | C | T | C | 0.125 | 0.127 | 0.120 | 0.121 |
| H5 | C | T | G | G | C | C | G | C | 0.081 | 0.073 | 0.085 | 0.075 |
| H6 | T | T | T | G | C | C | G | C | 0.024 | 0.041 | 0.024 | 0.051 |
| H7 | C | T | G | A | A | C | T | C | 0.023 | 0.017 | 0.031 | 0.021 |
| H8 | C | T | T | G | G | C | T | C | 0.013 | 0.018 | 0.011 | 0.017 |
| H9 | C | T | T | G | C | T | G | C | 0.006 | 0.020 | 0.009 | 0.021 |

Most common haplotype CTTGCCGC (H1) that carries the disease risk allele of the proxy for Y402H (1.2) used as reference
H1–H9 account for 98.5 of 2012 chromosomes in Cambridge and EUGENDA

FHR-4 levels; this could be achieved using Abs or other agents that block or sequester the protein or by anti-sense targeting of hepatic FHR-4 synthesis. The efficacy of clinical trials evaluating FHR-4 inhibiting treatments could be enhanced by patient selection based on FHR-4 levels and the genetic markers identified here.

## Methods

**Study samples.** The Cambridge AMD study is a case–control study with subjects recruited from the southeast and northwest of England between 2002 and 2006[65]. All affected subjects had CNV and/or GA. Controls were spouses, partners or friends of index patients. Blood samples were obtained at the time of interview; EDTA and lithium–heparin plasma samples were used for DNA extraction and FHR-4/FH measurements, respectively. The EUGENDA created for clinical and

**Fig. 5 Haplotype analysis identifies *CFH* locus haplotypes strongly associated with AMD and FHR-4 levels. a** Illustrates the association of the observed common nine haplotypes formed by the seven AMD-associated *CFH* locus variants considered in our association analyses and rs6677604 (overall haplotype frequency ≥1% in the Cambridge and EUGENDA cohorts combined, accounting for 98.5% of 2012 chromosomes) with AMD and with FHR-4/ FH levels. Details of the alleles forming the haplotypes together with the frequency distribution in the two cohorts combined, and as estimated in the IAMDGC dataset[5] (16,144 patients with advanced AMD and 17,832 controls of European ancestry), are shown in **b**: haplotype CTTGCCGC (H1) that carries the AMD increasing-risk allele T of the proxy for Y402H (1.2) is used as reference (coloured in red); alleles that are different from the reference are coloured in blue; the direction of association with AMD for the minor allele of each single variant as estimated in the IAMDGC study[5] is indicated within parentheses. Four association plots are displayed in **a**: the first two (top) plots show the OR (with CI) estimates for the *CFH* haplotype association with AMD in the IAMDGC dataset and our two-cohort meta-analysis, respectively; the third and fourth (bottom) plots show the β (with CI) estimates for the *CFH* haplotype association with FHR-4 and FH levels, respectively, in our two-cohort meta-analysis; haplotype H1 is used as reference. The estimates shown in each plot are labelled further to indicate the presence of alleles that differ from the reference; those alleles are indicated with the IAMDGC association signal numbers of the corresponding variants (1.1, 1.5–1.7), in red to indicate that the allele different from the reference is AMD risk-increasing allele, in blue if protective; the Y402H label is blue to indicate the presence of the protective allele G of variant 1.2, red for the AMD risk-increasing allele T; finally, the label DEL indicates the presence of the protective allele A of the proxy for the *CFHR1-3* deletion (rs6677604). See Supplementary Data 9 for full details of the haplotype association estimates. Source data are provided as a Source Data file.

molecular analysis of AMD comprises late AMD cases and controls recruited at Radboud University Medical Centre, the Netherlands, and University of Cologne, Germany. Details on exclusion criteria and grading are provided in the Supplementary Methods. All participants provided written informed consent for clinical examination, epidemiological data collection and blood sampling for biochemical and genetic analyses. Serum and plasma samples were used for FHR-4/FH measurements. Donor eye tissue was obtained from Manchester Eye Tissue Repository (ethically approved Research Tissue Bank, UK NHS Health Research Authority ref. 15/NW/0932). The banked tissue was collected and stored within 48 h of death; there was prior informed consent for research use. Human Tissue Act 2004 (UK) guidelines were followed. For all studies, ethical approval was obtained from either national or local ethics committees (NRES Committee East Midlands – Derby for the Cambridge AMD study; Arnhem–Nijmegen Commissie Mensgebonden Onderzoek (CMO) and Ethics Commission of Cologne University's Faculty of Medicine for EUGENDA) and adhered to the tenets of the Declaration of Helsinki.

**Proteins and Abs**. Recombinant FHR-4 was made through the GenScript gene synthesis and protein expression service (Piscataway, NJ, USA) using their baculovirus-insect cell expression system and was based on the published sequence for the FHR-4B variant of the *CFHR* gene (UniProt identifier Q92496-3): the protein was designed to include an N-terminal 6×His tag and TEV cleavage site (Supplementary Fig. 9).

For the generation of specific FHR-4 mAbs, mice were immunized subcutaneously with recombinant FHR-4 (~30 μg/mouse) in complete Freund's adjuvant; boosted 4 and 6 weeks later with FHR-4 (dose as above) in incomplete Freund's adjuvant and test bled at 8 weeks. Mice with the highest titre in a screening assay on immobilized FHR-4 protein were selected and boosted intraperitoneally with FHR-4 (30 μg in phosphate-buffered saline (PBS)), sacrificed 48 h later and the spleen harvested aseptically. Spleen cells, obtained by perfusion with RPMI in a sterile cabinet, were fused with SP2 myeloma cells to generate hybridomas using standard protocols. Cells were plated at limiting dilution in 96-well plates and left undisturbed for 14 days. Supernatant (50 μl) was removed from each well and screened for anti-FHR-4 titre as above. Positive clones were subjected to three rounds of re-cloning prior to expansion and large-scale culture. Abs were purified on protein G and tested in Western blotting against recombinant FHR-4 and human serum. Non-competitive pairs of Abs were identified for enzyme-linked immunosorbent assay (ELISA) development.

Recombinant FHL-1 was expressed in HEK293 cells as described previously[15]. Commercially available purified complement proteins used include C3b (VWR International, Lutterworth, UK, catalogue no. 204860), FH (Sigma-Aldrich, catalogue no. C5813), and FI (VWR International, catalogue no. 341280). Commercially available Ab against collagen IV was used (catalogue no: 600-401-106S, 2B Scientific Ltd., Oxford, UK).

**FHR-4 and FH systemic level measurements**. The levels of FHR-4 were measured using an optimized in-house sandwich ELISA. Nunc-Immuno™ MaxiSorp™ 96-well plates were coated with 50 μl per well of monoclonal anti-FHR-4 Ab 4E9 at 5 μg ml⁻¹ (in 0.1 M carbonate buffer pH 9.6). After blocking in 2% bovine serum albumin (BSA) in PBS + 0.1%Tween-20 (PBST), plates were washed in PBST and a dilution series of purified FHR-4 protein diluted in 0.1% PBST added to wells in duplicate to generate a standard curve. Test samples were added (50 μl per well) in duplicate at a 1:40 dilution to the remaining wells, and plates were incubated at 37 °C for 1.5 h. Plates were washed in PBST, 50 μl per well of 1 μg ml⁻¹ of horse radish peroxidase (HRP)-labelled anti-FHR-4 mAb clone 17 was added and the plates were incubated for 1 h at room temperature. After washing, 50 μl per well of orthophenylenediamine (SIGMAFAST™ OPD, Sigma-Aldrich, UK) was added to develop the plates and the reaction was stopped after 5 min by adding an equal volume of 10% sulfuric acid. Absorbance was measured in a plate reader at 492 nm

and protein concentrations were interpolated from the standard curve plotted using GraphPad Prism 5.

FH levels were measured in a similar manner using monoclonal anti-FH Ab OX24 at 5 μg ml⁻¹ as capture, purified FH protein diluted in 0.1% PBST as standard, test samples at a 1:4000 dilution, HRP-labelled monoclonal anti-FH Ab 35H9 (1 μg ml⁻¹) as the detection antibody, developed with OPD and read as above.

**Immunohistochemistry**. Human donor eye tissue sections were obtained from the Manchester Eye Tissue repository where 5 mm biopsies of the macula region from donor eyes were frozen in OCT and undergone cryo-sectioning (10 μm) that were subsequently stored at −80 °C. Frozen tissue section slides were stained for the presence of FHR-4, collagen IV or C3/C3b using methods described previously[15]. Briefly, tissue sections were incubated with chilled (−20 °C) histological grade acetone:methanol (1:1, v/v; Sigma-Aldrich) for 20 s before thorough washing with PBS. Tissue sections were blocked with 0.1% (w/v) BSA, 1% (v/v) goat serum, and 0.1% (v/v) Triton X-100 in PBS for 1 h at room temperature. After washing, tissue sections were incubated with Ab combinations of either 10 μg ml⁻¹ of anti-FHR-4 mAb (clone 150) mixed with either 1 μg ml⁻¹ anti-Collagen IV rabbit polyclonal Ab, or 1 μg ml⁻¹ anti-C3/C3b rabbit polyclonal antibody (catalogue no: 21337-1-AP, Proteintech Group Inc., United States), for 16 h at 4 °C. Sections were washed and biotinylated anti-mouse immunoglobulin G (IgG) (catalogue no. BA_9200, Vector laboratories Inc.) diluted 1:250 in PBS was applied for 1 h to amplify the FHR-4 signal. Slides were subsequently washed and Alexa Fluor® 647 streptavidin (catalogue no: S32357, Invitrogen) diluted 1:250 in PBS and Alexa Fluor® 488-conjugated goat anti-rabbit Ab (Invitrogen, USA) diluted 1:500 in PBS were added for 2 h at room temperature. After washing, DAPI (4′,6-diamidino-2-phenylindole) was applied as a nuclear counterstain (at 0.3 mM for 5 min) prior to mounting with medium (Vectashield; H-1400, Vector Laboratories, Peterborough, UK) and application of a coverslip.

In the case of blank control sections, an identical protocol was followed, but PBS replaced the primary antibody. To test antibody specificity in immunohistochemistry, pre-adsorption experiments were performed whereby 10-fold molar excess of recombinant FHR-4 was premixed with the anti-FHR-4 mAb prior to application to the tissue sections (Supplementary Fig. 10). Further testing was performed by pre-absorbing with excess purified FHL-1 protein to ensure the anti-FHR-4 antibody did not cross-react (Supplementary Fig. 10). Furthermore, competition ELISAs were performed demonstrating the specificity of clone 150 for FHR-4 and not FH (Supplementary Fig. 11). In all cases images were collected on a Zeiss AxioImager D2 upright microscope using a ×40/0.5 EC Plan-neofluar and ×100/0.5 EC Plan-neofluar objective and captured using a Coolsnap HQ2 camera (Photometrics) through Micromanager software v1.4.23. Specific band pass filter sets for DAPI, FITC and Cy5 were used to prevent bleed through from one channel to the next. Images were then processed and analysed using Fiji ImageJ (http://imagej.net/Fiji/Download).

**Surface plasmon resonance**. The binding of FHR-4 to immobilized C3b was measured by surface plasmon resonance using a Biacore 3000 (GE Healthcare). The sensor surfaces were prepared by immobilizing human C3b onto the flow cells of a Biacore series S carboxymethylated dextran (CM5) sensor chip (GE Healthcare) using standard amine coupling and included blank flow cells where no C3b protein was present. Experiments were performed at 25 °C and a flow rate of 15 μl min⁻¹ in PBS with 0.05% surfactant P20. FHR-4 was injected in triplicate at concentrations ranging from 1 to 100 μg ml⁻¹. Samples were injected for 150 s and dissociated for another 200 s; the chip was regenerated with 1 M NaCl for 1 min and re-equilibrated into PBS with 0.05% surfactant P20 prior to the next injection. After subtraction of the blank cell value from each response value, association and dissociation rate constants were determined by global data analysis. All curves were

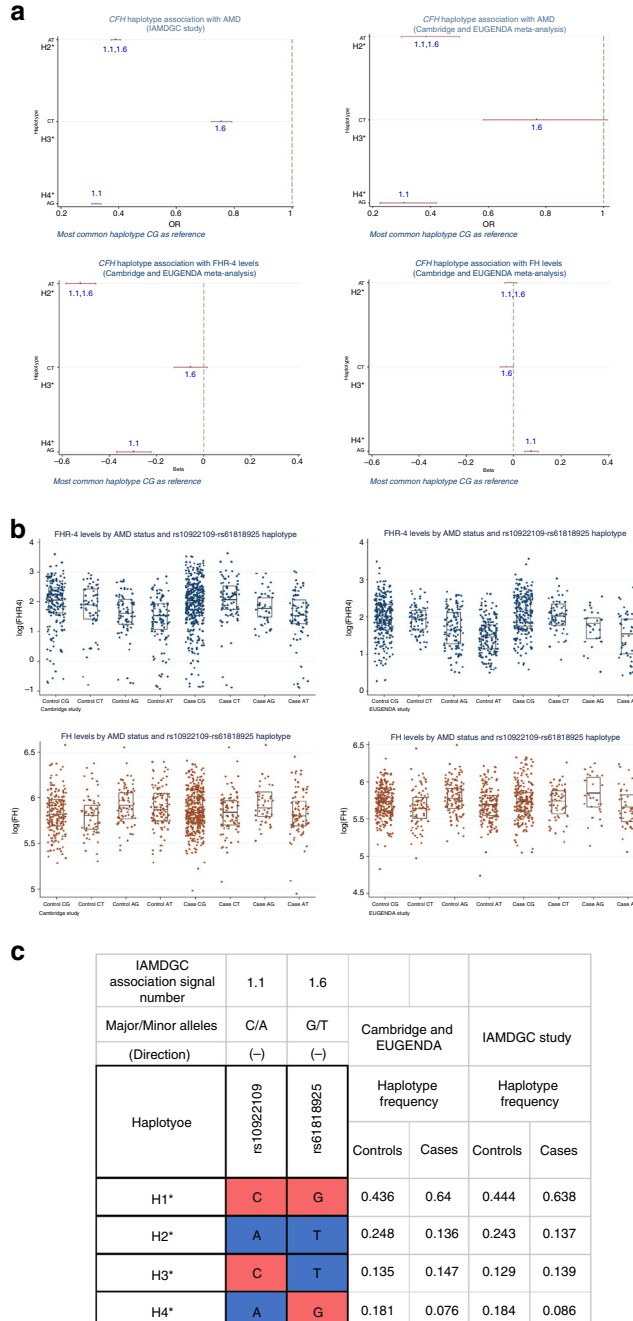

**Fig. 6 The AMD-associated variants rs10922109 (1.1) and rs61818925 (1.6) are a minimal set of variants that explain the genetic effect on FHR-4 levels at the *CFH* locus. a** illustrates the association of the observed four haplotypes formed by the two independently AMD-associated variants rs10922109 (1.1) and rs61818925 (1.6) at the *CFH* locus with AMD and with FHR-4/FH levels. Variants 1.1 and 1.6 represent the best two single-variant association signals with FHR-4 levels in the Cambridge and EUGENDA meta-analysis (Table 2). Details of the alleles forming the haplotypes together with the frequency distribution in the two cohorts combined (484 patients with advanced AMD and 522 controls) and as estimated in the IAMDGC dataset[5] (16,144 patients with advanced AMD and 17,832 controls of European ancestry) are shown in **c**: most common haplotype CG (H1*) is used as reference (coloured in red); alleles that are different from the reference are coloured in blue; the direction of association with AMD for the minor allele of each single variant as estimated in the IAMDGC study[5] is indicated within parentheses. Four association plots are displayed in **a**: the first two (top) plots show the OR (with CI) estimates for the *CFH* haplotype association with AMD in the IAMDGC dataset and our two-cohort meta-analysis, respectively; the third and fourth (bottom) plots show the β (with CI) estimates for the *CFH* haplotype association with FHR-4 and FH levels, respectively, in our two-cohort meta-analysis; haplotype H1* is used as reference. The estimates shown in each plot are labelled further to indicate the presence of alleles that differ from the reference; those alleles are indicated with the IAMDGC association signal numbers of the corresponding variants (1.1 and 1.6), in blue to indicate that the allele different from the reference is AMD protective. See Supplementary Data 11 for full details of the rs10922109–rs61818925 haplotype association estimates. Finally, **b** shows box plots (and corresponding data points) of FHR-4 levels (top) and FH levels (bottom) by rs10922109–rs61818925 haplotype for each study cohort (Cambridge and EUGENDA). Each box plot depicts median value (central line), first quartile (lower bound line) and third quartile (upper bound line). Source data are provided as a Source Data file.

**c**

| IAMDGC association signal number | 1.1 | 1.6 | | | | |
|---|---|---|---|---|---|---|
| Major/Minor alleles | C/A | G/T | Cambridge and EUGENDA | | IAMDGC study | |
| (Direction) | (−) | (−) | | | | |
| Haplotype | rs10922109 | rs61818925 | Haplotype frequency | | Haplotype frequency | |
| | | | Controls | Cases | Controls | Cases |
| H1* | C | G | 0.436 | 0.64 | 0.444 | 0.638 |
| H2* | A | T | 0.248 | 0.136 | 0.243 | 0.137 |
| H3* | C | T | 0.135 | 0.147 | 0.129 | 0.139 |
| H4* | A | G | 0.181 | 0.076 | 0.184 | 0.086 |

Most common haplotype CG (H1*) used as reference

fitted using a 1:1 Langmuir association/dissociation model (BIAevaluation 4.1; GE Healthcare).

**Solid-phase binding assays**. Purified C3b was adsorbed onto the wells of microtiter plates (Nunc Maxisorb, Kastrup, Denmark) at 1 μg per well in 100 μl per well PBS for 16 h at room temperature. Plates were blocked for 90 min at 37 °C with 300 μl per well 1% (w/v) BSA in standard assay buffer (SAB; 20 mM HEPES, 130 mM NaCl, 0.05% (v/v) Tween-20, pH 7.3). SAB was used for all subsequent incubations, dilutions and washes and all steps were performed at room temperature. A constant concentration of 100 nM was made for either FH or FHL-1 in SAB and increasing concentrations of FHR-4 are used as competitor, up to 500 nM. FH/FHR-4 and FHL-1/FHR-4 mixes were incubated with the immobilized C3b for 4 h. After washing, bound FH or FHL-1 protein was detected by the addition of 100 μl per well of 0.5 μg ml⁻¹ OX23 antibody and incubated for 30 min, followed by washing and a 30-min incubation in 100 μl of a 1:1000 dilution of AP-conjugated anti-mouse IgG (Sigma-Aldrich). Plates were developed using 100 μl per well of a 1 mg ml⁻¹ disodium *p*-nitrophenylphosphate solution (Sigma-Aldrich) in 0.05 M Tris-HCl, 0.1 M NaCl, pH 9.3. The absorbance values at 405 nm

were determined after 10 min of development at room temperature and corrected against blank wells (i.e., those with no immobilized C3b).

**Fluid-phase C3b breakdown assays**. The fluid-phase cofactor activity of FHL-1 was measured by incubating purified FHL-1, C3b and FI together in a total volume of 20 μl PBS for 15 min at 37 °C. For each reaction 2 μg C3b and 0.04 μg FI were used with varying concentrations of FHL-1 ranging from 0.015 μg to 1 μg per reaction. The assay was stopped by the addition of 5 μl 5× sodium dodecyl sulfate (SDS) reducing sample buffer and boiling for 10 min at 100 °C. Samples were run on a 4–12% NuPAGE Bis Tris gel at 200 V for 60 min in order to maximize the separation of the C3b breakdown product bands (Supplementary Fig. 4). Molecular weight markers used were Novex Sharp pre-stained protein standards (3.5–260 kDa, catalogue no. LC5800, Life Technologies, Paisley, UK). The density of the 68 kDa iC3b product band was measured using ImageJ64 (version 1.40g; rsb.info.nih.gov/ij) and used to track C3b breakdown efficiency of the FHL-1 proteins. For FHR-4 inhibition assays, the amount of FHL-1 used in the reaction is fixed at 1 μg and increasing amounts of FHR-4 were added to create up to a 5-fold molar excess of FHR-4 over FHL-1. Otherwise, the reactions were performed under the same condition as previously. In all cases averaged data from three separate experiments were used.

**Ussing chamber diffusion experiments**. The macular region of enriched BrM isolated from donor eyes was mounted in an Ussing chamber (Harvard Apparatus, Hamden, CT)[61]. Once mounted, the 5-mm-diameter macular area was the only barrier between two identical compartments (Supplementary Fig. 3). Both sides of BrM were washed with 2 ml PBS for 5 min at room temperature. Fresh PBS was placed in both the sample and diffusate chambers. To the sample chamber, pure recombinant FHR-4, final concentration of 100 μg ml⁻¹, was added and the Ussing chamber was left at room temperature for 24 h with gentle stirring in each compartment to avoid generating gradients of diffusing protein. Samples from each chamber were analysed on 4–12% NuPAGE Bis-Tris gels, run at 200 V for 60 min. Either 20 μl samples straight from each chamber were mixed with 5 μl 5× SDS loading buffer and run or 100 μl samples were taken and concentrated using StrataClean beads (hydroxylated silica; Agilent Technologies, Cheadle, UK) for 5 min at room temperature before centrifugation. Beads were then re-suspended in 20 μl neat 5× SDS loading buffer and loaded directly to the gel. Gels were stained

with Instant Blue stain (Expedeon, Harston, UK) for 60 min at room temperature, before washing and storage in MiliQ water. Molecular weight markers used were Blue Prestained Protein Standards, Broad Range (11–190 kDa, New England Bio-Labs, Hitchin, UK, catalogue no. P7706S). Diffusion experiments were performed on three separate donor BrM.

**Genotype data and association analysis**. All individuals included in this study had been previously genotyped with a custom-modified Illumina HumanCoreExome array at the Centre for Inherited Disease Research (CIDR) and analysed within the IAMDGC GWAS (43,566 subjects; 16,144 advanced AMD cases and 17,832 controls of European ancestry in the primary analysis dataset)[5]. Quality control and genotype imputation using the 1000 Genomes Project[66] reference panel were performed by the IAMDGC as described previously[5]. A total of 9,618,989 quality-controlled common (minor allele frequency (MAF) ≥1%) variants (289,971 genotyped; 9,329,018 imputed) were available for the 1006 individuals included in this study. Phased genotype data as inferred within the IAMDGC study[5] were also available and used in the haplotype-based association analyses. All statistical association analyses were conducted on each cohort separately (Cambridge and EUGENDA), and combined as two-stage, fixed-effects meta-analyses of the available individual participant data from the two cohorts. Heterogeneity across studies was assessed using the $I^2$ statistic. FHR-4 and FH levels were natural logarithmically transformed to ensure normality of the distribution when using linear regression models. We assessed the association of late AMD with natural logarithmically transformed FHR-4/FH levels via Wald tests using linear regression models adjusted for sex, age, batch effects and the first two genetic principal components (as estimated within the IAMDGC study[5]). We also reported the association of FHR-4/FH levels with late AMD via OR expressed as per standard deviation (SD) change of log levels using logistic regression models adjusted for sex, age, batch effects and the first two genetic principal components. We assessed the association of the eight independently AMD-associated variants at the *CFH* locus reported by the IAMDGC study[5] (i.e., rs10922109 [1.1], rs570618 [1.2], rs121913059 [1.3], rs148553336 [1.4], rs187328863 [1.5], rs61818925 [1.6], rs35292876 [1.7], rs191281603 [1.8]; Supplementary Data 1) with natural logarithmically transformed FHR-4/FH levels via Wald tests on the variant genotypes coded as 0, 1 and 2 according to the number of minor alleles for the directly typed variants or allele dosages for the imputed variants, using linear regression models adjusted for sex, age, batch effects and the first two genetic principal components in controls, and in all samples further adjusting for AMD status. The single-SNP association with AMD was assessed with ORs expressed as per 1 minor allele, using logistic regression models adjusted for the first two genetic principal components. Finally, we extracted the best-guess (i.e., most likely) haplotypes formed by the AMD-associated variants at the *CFH* locus considered in our analysis and additionally included rs6677604 as proxy for the AMD-protective *CFHR1–3* deletion[29], using the phased genotype data produced within the IAMDGC study[5]. The association of the observed haplotypes with AMD was assessed using logistic regression models adjusted for the first two genetic principal components, and with FHR-4/FH levels using linear regression models adjusted for AMD status, sex, age, batch effects, and the first two genetic principal components. The haplotype-based association with AMD was also performed on the whole IAMDGC primary analysis dataset of 16,144 patients with advanced AMD and 17,832 control subjects of European ancestry using logistic regression models adjusted for whole-genome amplification and the first two genetic principal components as per the IAMDGC study[5]. All the statistical analyses above were conducted using the Stata software, version 13.1 (StataCorp); *tobit* command was used for censored regression models to take into account any 'below of detection' FHR-4 levels ($n = 16$ data points equal to baseline 0.504116; with virtually identical results as per *regress* command for linear regression models); *ipdmetan* and *mvmeta* commands were used for conducting meta-analyses of individual participant data.

We also carried out GWASs of natural logarithmically transformed FHR-4 and FH levels in controls from each cohort (Cambridge and EUGENDA) using linear regression models adjusted for sex, age, batch effects and the first two genetic principal components, and in all samples further adjusting for AMD status. The GWASs were carried out using the EPACTS software (http://genome.sph.umich.edu/wiki/EPACTS) and Wald tests were performed on the variant genotypes coded as 0, 1 and 2 according to the number of minor alleles for the directly typed variants or allele dosages for the imputed variants. Genomic control correction[67] was applied if λ was >1. Effect size estimates and standard errors of single variants seen in both cohorts were subsequently combined in a fixed-effect meta-analysis using METAL[68]. This meta-analysis had a statistical power of over 80% to detect associations of genetic variants with a MAF ≥1% explaining ≥3.9% of the variance in FHR-4 levels (Genetic Power Calculator: http://zzz.bwh.harvard.edu/gpc/). Manhattan and Q–Q plots were generated using the *qqman* R package (version 0.1.2). Regional plots of association were generated using LocusZoom (version v0.4.8)[69]. Finally, LD measures ($R^2$ and $D'$) were calculated using LDlink (https://ldlink.nci.nih.gov/), based on the European (EUR) population genotype data originates from Phase 3 (Version 5) of the 1000 Genomes Project[66].

**Reporting summary**. Further information on research design is available in the Nature Research Reporting Summary linked to this article.

## Data availability
The summary statistics for the GWAS meta-analyses of FHR-4 and FH levels are available through the UCL Research Data Repository at https://rdr.ucl.ac.uk/ [https://doi.org/10.5522/04/11396565].

The Gene Expression Omnibus datasets used for the gene expression analyses are: GSE18811; GSE41102; GSE50195; GSE94437; GSE99248. The Genotype-Tissue Expression (GTEx) Project datasets used for the gene expression analyses were obtained from the GTEx Portal, https://gtexportal.org/home/multiGeneQueryPage (4/4/2018), dataset dbGaP accession number phs000424.v8.p2; the GTEx Project was supported by the Common Fund of the Office of the Director of the National Institutes of Health, and by NCI, NHGRI, NHLBI, NIDA, NIMH and NINDS. The source data underlying Figs. 1, 2b–i, 3b, 4, 5a, 6a, b and Supplementary Figs. 2a, 5, 6, 7a–d, 8, 10a, 11 are provided as a Source Data file. All other datasets and reagents generated/used in the current study are available from the corresponding authors upon reasonable request.

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

## Acknowledgements

We are grateful to all the subjects who kindly participated in this research. For the Cambridge AMD Study (UK Medical Research Council (MRC) grant G0000067 to J.R.W.Y. and A.T.M.), we gratefully acknowledge help with patient recruitment from members of the Genetic Factors in AMD Study Group (P. Black, Z. Butt, V. Chong, C. Edelsten, A. Fitt, D.W. Flanagan, A. Glenn, S.P. Harding, C. Jakeman, C. Jones, R.J. Lamb, V. Moffatt, C.M. Moorman, R.J. Pushpanathan, E. Redmond, T. Rimmer and D.A. Thurlby); we thank Jane Khan and Humma Shahid for carrying out the clinical evaluation and sampling of subjects and Tunde Peto and colleagues at the Reading Centre, Moorfields Eye Hospital, London, for grading the fundus photographs. EUGENDA was funded by grants from the Oogfonds, MaculaFonds, Landelijke Stichting voor Blinden en Slechtzienden, Stichting Blindenhulp, Stichting A.F. Deutman Oogheelkunde Researchfonds, the Netherlands Organization for Scientific Research (Vidi Innovational Research Award 016.096.309), and the European Research Council under the European Union's Seventh Framework Programme (FP/2007-2013) (ERC Grant Agreement no. 310644 MACULA). We wish to thank the Manchester Eye Tissue Repository for supplying human macula tissue, and their funding from the Macular Society UK (12928). We also thank the IAMDGC (http://eaglep.case.edu/iamdgc_web/; for a full list of consortium

members, please see Supplementary Note 1) for providing the genotype data. The Cambridge and EUGENDA samples were genotyped as part of the IAMDGC exome-chip project supported by CIDR (contract number HHSN268201200008I) and funded by EY022310 (to Jonathan L. Haines, Case Western Reserve University, Cleveland) and 1 × 01HG006934-01 (to Gonçalo R. Abecasis, University of Michigan, Department of Biostatistics). We wish to thank Lars Fritsche for providing *CFH* locus-phased genotype data for the IAMDGC primary analysis dataset. Other funding sources are as follows: S.J.C./V.T., an MRC fellowship (MR/K024418/1); S.M., Fight for Sight UK research grant (1517/1518); B.P.M./D.F. supported by a Programme Grant from the MRC-supported UK Dementia Research Institute; V.C. was primarily funded by the Department of Health's NIHR Biomedical Research Centre for Ophthalmology at Moorfields Eye Hospital and UCL Institute of Ophthalmology, and an MRC research grant (MR/P025838/1). The funding bodies had no role in the design of the study and collection, analysis, and interpretation of data and in writing the manuscript.

## Author contributions

V.C. and L.L.-M. performed the statistical association analyses and primarily wrote the manuscript. F.H. performed IHC for FHR-4, collagen IV and C3b on human macular tissue sections. D.F. performed FH and FHR-4 blood level analysis in both Cambridge and EUGENDA cohorts. V.T. made recombinant FHL-1 used in biochemical experiments and FHR-4 used to generate anti-FHR-4 antibody. S.M. collected and processed human eye tissue for the study. N.B. re-analysed gene expression data from the public data repository's Gene Expression Omnibus and Expression Atlas. İ.E.A. helped with the GWAS meta-analyses of FHR-4 and FH levels. A.T.M. and J.R.W.Y. are principal investigators for the Cambridge AMD study, collected patient blood samples and clinical and genetic data. C.B.H., S.F., E.K.d.J. and A.I.d.H. collected patient blood samples and are custodians of the EUGENDA sample cohort. B.P.M. generated anti-FHR-4 mAbs, designed and optimized the FH- and FHR-4 specific ELISA and contributed to the primary writing of the manuscript. P.N.B. contributed to the design of experiments, to collection of Cambridge AMD study samples, writing of the manuscript and supervised IHC experiments. S.J.C. coordinated the project and performed biochemical analysis, including binding, competition and C3b breakdown assays with FHR-4, designed IHC experiments, and contributed to the primary writing of the manuscript. All authors contributed to data interpretation and the final version of the manuscript text.

## Competing interests

The authors declare no competing interests.
