## [Peer Review File · Nature Communications]

Reviewers' comments:

Reviewer #1 (Remarks to the Author):

The manuscript describes an extremely striking finding in the research field for age-related macular degeneration. In summary, the authors showed very clear elevation of FHR-4 levels in serum for AMD cases compared to controls in 2 independent collections (Cambridge; 304 cases and 214 controls and EUGENDA; 180 cases and 308 controls). Contrastingly, no such difference between AMD cases and controls was observed for serum FH levels. The authors performed a series of biological experiments which allowed them to draw an inference regarding a potential mechanistic link between increased FHR-4 competing with FH inactivation of the C3 pathway, leading to activation of complement and increasing risk of AMD (Figure 2, main text). Crucially, there is already strong support from prior work (Clark SJ et al., *J Immunol.* 2014; 193(10): 4962–4970) showing that it is not unexpected that serum FH levels itself is not associated with AMD risk. This is because the FH protein cannot get into the Bruch membrane whereas the smaller FHL-1 is able to. The data is clearly and convincingly presented.

Minor points:

1. In Table 1, the comparison of FH and FHR-4 serum levels between AMD cases and controls was undertaken using linear regression, adjusting for sex, age, and batch effects. Could the authors also consider adding in the top 2 principal components of genetic stratification into both models? This will further exclude a possibility that genetic stratification could explain the differences in FHR-4 levels between AMD cases and controls.
2. It has been very clearly established that the very rare CFH R1210C variant showed an extremely big effect (OR > 30 on a fully adjusted model by Fritsche LG et al., *Nat Genet* 2016; 48:134-43). The effect of this protein-changing mutation in CFH (which encodes for FH) cannot be denied, yet its effect was not discussed. Could the authors discuss how the risk effect of CFH R1210C square with the abrogation of the protective effect of FHL-1 (is R1210C part of the FHL-1 protein?), knowing that FHL-1 is a truncated form of FH? I could not clarify this by referring to Clark et al., 2014.
3. I earnestly feel that the authors have under-sold the significance of Figure 4 (main text). Granted, the association with FHR-4 is extremely significant. However, the authors also note non-trivial genome-wide significant association with FH as well, in the same CFH locus region with common variants. I think the data is trying to teach us something regarding the balance between FH and FHR-4 and AMD disease biology. I feel that the emphasis has been too much on FHR-4, with too little regard for FH itself despite the weight of evidence shown by the data. What do the authors think?

Reviewer #2 (Remarks to the Author):

Overall

The authors state they demonstrate a key role for FHR-4 in regulating cofactor activity in the human eye in AMD. The most important aspect of this paper is the evidence suggesting that FHR-4 competes in the eye with FH and FHL-1 to control complement activation.

Employing a combination of genetics, antigenic/functional analyses and retinal microscopy of FHR-4 makes for an interesting story. However, there are some questions and comments below.

General Comments

1. Anything that causes a decrease in complement activation for C3b in the eye predisposes to AMD. This is the message from the literature. Haplo-insufficiency of FI or FH secondary to rare genetic variants is highly predisposing to AMD. If the variant is in CCP repeats 1-7, then FHL-1 also carries the mutation. Albeit possibly overstating their case, the authors now show a role for FHR-4, supporting some prior genetic data on deletion of FHR-1 and FHR-4 as being associated with reduced AMD risk. Of note, however, the largest GWAS to date shows that CFHR3/CFHR1 does not reduce risk (conditioned analysis shows OR =1.18).

2. The authors take advantage of a specific mAb to FHR-4 that they have developed. They are to be congratulated for making this mAb.

3. In addition to the highly penetrant R1210C story (obviously doesn't directly involve FHL-1), several other familial forms of AMD have been identified with variants in FH unrelated to FHL-1, i.e. the major problem in those AMD cases is decreased FH antigenic levels or normal expression quantities but of dysfunctional protein – again leading to decreased FH function. So FHR-4 levels do not drive disease in those cases. Also, the known FI story tells us that cofactor activity for C3b is the key regulatory event in the eye to control complement activation. Haplo-insufficiency of FI is highly penetrant in AMD. The authors do not discuss or adequately reference these cases.

Specific Comments;

1. The FHR-4 levels in Figure 1A do not seem to match the FHR-4 levels shown in table 1. Please clarify and explain. For example, mean level is 5.5 (Table 1) but looks like 7 or 8 in Figure 1A. Note the overlap between controls from 15 and below with AMD – very few cases are not overlapping especially in the series Fig 1B. Also, the dark blue color obscures the black linear lines.

2. Figure 2. FH is more difficult to compete off of C3b by increasing concentration of FHR-4 than FHL-1. In fact, between approx. 100 and 500 nM there is not a strong Linear FH inhibition profile relative to binding to C3b. Are the concentrations of FHR-4 employed approximately what the authors find in the eye? Figures H and I : only show competing against FHL-1, not FH. Are these concentrations in the ballpark that one sees in the retina? Figure J does not show FH. Figures B-D also do not show FH and FHL.

4. Why was the FHR1-4 deletion not included in the authors' analysis? It was present in 3 controls and 1 AMD patient. Also, CFH R1210C was not included in the analysis. Why? Do the results go against their hypothesis?

5. Why didn't the two protective haplotypes (carrying FHR-4 lowering alleles) associate with AMD?

6. The first sentence in the Discussion is not accurate. Low antigenic/functional levels of FH,

of course, are associated with AMD – the rare variant story is independent of FHR-4.

7. The authors often fail to note a key role for FH or FI, focusing instead on FHR-4 and FHR

1. There are variants outside of FH repeats 1-7. Explain?

Other:

Table 1 is difficult to navigate. Looks like 2 tables merged together. The four figures are actually about 20 figures merged together.

Reviewer #3 (Remarks to the Author):

This is a very interesting report that, if proved accurate, will change the way the field thinks about macular degeneration genetic risk, as well as elevating a humble member of the complement regulator family to prominence.

The authors generated a set of monoclonal antibodies directed against the factor H related protein, FHR-4. This antibody was then used in ELISA assays to assess circulating levels of FHR-4 in the blood of two independent, large cohorts of either controls or AMD patients (with advanced AMD, including both GA and CNV patients).

They found that the levels of FHR-4, while low, were elevated about 20% on average in the late AMD cases. This difference was statistically significant. In contrast, levels of fH were not altered between cases and controls.

When the samples are stratified by genotypes at the 1q31 locus, as opposed to affection status, a much more powerful striking and association with elevated FHR-4 was identified. This single variant association holds true when patients are stratified using haplotype blocks. When subjected to genome wide analysis, the only region in the genome associated with elevated FHR-4 at genome wide significance is 1q31.

Functionally, the authors demonstrate binding of an anti-FHR-4 antibody on tissue sections in the inter-capillary pillars of the choroid and incompletely wrapping around the capillaries and in a hard druse. They further demonstrate using surface plasmon resonance that FHR-4 binds C3b and that, at very high concentrations, can inhibit the breakdown of C3b.

This is an exciting and overall potentially very important study and is of interest to readers of Nat Comm.

The main concern this reviewer has regards the specificity of the reagents. Since the entire study depends completely on the quality and specificity of the FHR-4 antibody, the convincingness of the antibody must be very solid and further characterization is necessary.

The supplemental data add some helpful insight: (Figure S8). In S8A, a large excess of antigen (the same recombinant protein used to generate the MAb) is able to compete away the binding on tissue sections. This may be slightly circular since the same antigen can block the antiserum, but it seems reasonable. In S8B, a reaction of a 75kDa band is shown from

serum. Preincubating the antibody with FHL-1 does not block labeling with the FHR-4 antibody, but how about preincubation with full length FH? Do the authors know where the epitope for Clone 150 resides? What if it corresponds to a domain of FH that's absent from FHL?

The question about specificity is also a concern because of the relatively low levels of FHR-4 detected in serum compared to FH. For FHR-4, the highest value (an outlier more than 3x higher than the mean) is 35ug/mL, whereas for FH the lowest outlier is about 180ug/mL. It would not be surprising if there is 100x more FH as FHR-4 in the choroid, especially as FH (unlike FHR-4) is abundantly synthesized in the eye. In light of this, there is concern that even though the MAbs may have much lower affinity for FH compared to FHR-4, the much more abundant FH protein may be responsible for the signal detected on IHC. This should ideally be demonstrated using protein biochemistry, especially precipitation of FH and FHR-4 from the RPE/Bruch's/choroid and MS of the pulled down fractions and minimally demonstration of FHR-4 and FH and FHL-1 levels in choroid protein extracts.

A few smaller issues to consider are listed below:

1. The conclusion in the abstract that FHR-4 is the key molecular player in AMD is overstated given the data.
2. Details on the patient and control cohort would be helpful. While the authors state that the statistics are adjusted for age, how much adjustment was necessary? If there was not overlap between the cases and controls, it may not be possible to correct for that with linear regression. For that matter, was FHR-4 level related to age at all in the dataset?
3. A cartoon of the difference between FH, FHL1 and FHR4 would be helpful to educate the reader and especially with the interpretation of the antibody and blocking experiments.
4. The authors say a few places that they used "phenotyped" donor tissue—does the phenotyping come into play otherwise in the manuscript?
5. The donor tissue is up to 48 hours after death. Can the authors convince the reader that this is an acceptable interval for IHC?
6. The quantitative aspects of the study are a little confusing. The authors note that a 2.5x molar excess of FHR-4 over FHL-1 caused 50% reduction in C3b cleavage. Is this a ratio that is likely to ever exist in nature in Bruch's/choroid? Again the biochemical demonstration of the abundance of these factors would be valuable, I would also suggest the authors include FH in the study.
7. Figure 3B- the authors are encouraged to please look at this figure carefully—the legends seem miniscule to this reader (although the data are great)—please remake this

8. It's a little concerning that advanced AMD is treated like a single entity. While the authors stratified based on genotype (and those data are very compelling) there is doubt as to whether CNV and GA should be grouped together. Can the authors split these and show if the signal is coming from one or both end stages?

9. The authors note that the circulating levels of FHR-4 are low but that the ratios of FHR-4 to FH/FHL-1 are probably different in serum than they are in the ECM of Bruch's. This is probably true--but it's not obvious that this ratio is tipped in a way that favors more FHR-4 on the ECM, especially with the high degree of local synthesis of FH in the RPE and the choroid. The authors are encouraged to determine this ratio with biochemistry.

10. One of the interesting implications from this work is that local synthesis in the eye of complement inhibitors is less important than the circulating regulators (i.e., FHR-4). This reviewer feels the jury is certainly out on that question, but wonders if the authors can address the recent studies by Khandhadia and colleagues that suggest liver transplantation genotype is less impactful on AMD risk compared to host somatic genotype.

Re. Nature Communications NCOMMS-19-08243-T

Note to all reviewers:

Many formatting changes have been applied to comply with the journal's guidelines; a new 150-word abstract has replaced the previous version and the required sections 'End notes' and 'Data availability' have been added; most of the methods that were previously in the Supplementary Materials are now included in the main text; many other edits are visible as track changes.

Reviewer #1:

The data is clearly and convincingly presented.

Reply: We thank the reviewer for their appreciation of the clarity and significance of our study.

1. In Table 1, the comparison of FH and FHR-4 serum levels between AMD cases and controls was undertaken using linear regression, adjusting for sex, age, and batch effects. Could the authors also consider adding in the top 2 principal components of genetic stratification into both models?

Reply: We have now adjusted our models further, adding the first two genetic principal components as suggested and estimates are identical to the first decimal point. We have also updated the corresponding adjusted P-values in Table 1, plots in Supplementary Figure 1 and corresponding main text in lines 121-122.

2. [...] Could the authors discuss how the risk effect of CFH R1210C square with the abrogation of the protective effect of FHL-1 (is R1210C part of the FHL-1 protein?), knowing that FHL-1 is a truncated form of FH?

Reply: The reviewer is correct to point out that the highly-penetrant R1210C variant is strongly associated with AMD and that this is not included in FHL-1. We have modified the final paragraph of the discussion (lines 298-304) to highlight this, and other clarifications requested from the reviewers.

3. I earnestly feel that the authors have under-sold the significance of Figure 4 (main text). [...] I feel that the emphasis has been too much on FHR-4, with too little regard for FH itself despite the weight of evidence shown by the data. What do the authors think?

Reply: This is now addressed in the modified final paragraph of the discussion (lines 298-315).

Reviewer #2:

1. Anything that causes a decrease in complement activation for C3b in the eye predisposes to AMD [...] Haplo-insufficiency of Fl or FH secondary to rare genetic variants is highly

predisposing to AMD. [...] Of note, however, the largest GWAS to date shows that CFHR3/CFHR1 does not reduce risk (conditioned analysis shows OR =1.18).

Reply: The reviewer is absolutely correct, that any factor that causes a decrease in complement regulation and/or increase in complement turnover is likely to contribute to complement-mediated AMD. To make this point clearer in our manuscript we have included additional text in the modified final paragraph of the discussion (lines 298-307).

2. The authors take advantage of a specific mAb to FHR-4 that they have developed. They are to be congratulated for making this mAb.

Reply: We thank the reviewer for their kind appraisal of our work.

3. [...] several other familial forms of AMD have been identified with variants in FH unrelated to FHL-1[...] So FHR-4 levels do not drive disease in those cases. Also, the known FI story tells us that cofactor activity for C3b is the key regulatory event in the eye to control complement activation. Haplo-insufficiency of FI is highly penetrant in AMD. The authors do not discuss or adequately reference these cases.

Reply: This is addressed in the modified final paragraph of the discussion (lines 298-307).

Specific Comments:

1. The FHR-4 levels in Figure 1A do not seem to match the FHR-4 levels shown in table 1. Please clarify and explain. For example, mean level is 5.5 (Table 1) but looks like 7 or 8 in Figure 1A. Note the overlap between controls from 15 and below with AMD – very few cases are not overlapping especially in the series Fig 1B. Also, the dark blue color obscures the black linear lines.

Reply: We apologise for the error, but the reviewer is perfectly correct! We had failed to update an older version of the table. A new Figure 1 has been produced (with same software and colouring as per other box-plot figures in the manuscript to ensure uniformity) and the values in new Figure 1 now match the ones presented in Table 1.

2. Figure 2. FH is more difficult to compete off of C3b by increasing concentration of FHR-4 than FHL-1. In fact, between approx. 100 and 500 nM there is not a strong Linear FH inhibition profile relative to binding to C3b. Are the concentrations of FHR-4 employed approximately what the authors find in the eye? Figures H and I : only show competing against FHL-1, not FH. Are these concentrations in the ballpark that one sees in the retina? Figure J does not show FH. Figures B-D also do not show FH and FHL.

Reply: Unfortunately, it is currently impossible to directly measure the levels of FHR-4 (and either FH or FHL-1) in the ECM of the choriocapillaris. This is because it is a highly vascular tissue and it is not possible to specifically isolate the extracellular matrix of the tissue to quantify its concentration of complement proteins. We infer that elevated systemic levels of FHR-4 will impact on the local levels in the extracellular matrix of the choriocapillaris and that protein ratios in blood are unlikely to represent ratios in the ECM (lines 289-295). FHL-1 and FH are not included in figure 2B-D simply because we are trying to show the localisation of FHR-4. The presence of FHL-1 in these tissues has been published previously (see Clark *et al.* (2014)) and can also be seen in our Supplementary Figure 10. We have modified Figure 2J to include FH and apologise for this oversight.

4. *Why was the FHR1-4 deletion not included in the authors' analysis? It was present in 3 controls and 1 AMD patient. Also, CFH R1210C was not included in the analysis. Why? Do the results go against their hypothesis?*

Reply: Both the *CFHR1-4* deletion and R1210C are rare. We state the corresponding absolute frequencies observed in our dataset in cases and controls that are consistent with the expected protective and deleterious effects (respectively), but we did not include these variants in the haplotype-based analyses as none of them is seen on any of the common haplotypes tested (with overall frequency $\geq 1\%$).

5. *Why didn't the two protective haplotypes (carrying FHR-4 lowering alleles) associate with AMD?*

Reply: The two haplotypes the reviewer refers to (H2: CTGGACTC and H3: CTGAACGC) are indeed strongly associated with lower risk of AMD (protective) using the most common haplotype CTTGCCGC (Y402H) as reference. In the main text we state: "Common H2-H5 and rarer H7 haplotypes carried significantly lower AMD risk than H1" (line 202). Full details of the corresponding association estimates are presented in Supplementary Data 9.

6. *The first sentence in the Discussion is not accurate. Low antigenic/functional levels of FH, of course, are associated with AMD – the rare variant story is independent of FHR-4.*

Reply: Respectfully, we believe the reviewer may have misunderstood our meaning. We state that AMD is associated with genetic-driven elevation of circulating FHR-4 and not FH, which is true. However, to avoid any doubt about the contribution of FH and non-FHR-4 mediated AMD we have clarified the point around genetic alterations to FH function and the risk of AMD by adding text later in the discussion (see lines 298-307).

7. *The authors often fail to note a key role for FH or FI, focusing instead on FHR-4 and FHR 1. There are variants outside of FH repeats 1-7. Explain?*

Reply: This is addressed in the modified final paragraph of the discussion.

Table 1 is difficult to navigate. Looks like 2 tables merged together. The four figures are actually about 20 figures merged together.

Reply: Multiple panel figures are accepted based on the journal's guidelines, but we apologise if some display items contained too much information. We have now split Table 1 into two different tables ("*Demographics of study cohorts and association analyses between AMD and systemic FHR-4/FH levels*" and "*Single-variant association analyses with FHR-4 and FH levels for the 8 AMD independently associated variants at the CFH locus from the IAMDGCC study*"). Out of the four figures originally provided, we agree that Figure 3 was particularly busy and difficult to interpret, yet containing a large amount of important findings. Figure 3 has now been split into three different figures (new Figure 3, Figure 5 and Figure 6) using original panels, and corresponding new figure legends have been added.

Reviewer #3:

This is a very interesting report that, if proved accurate, will change the way the field thinks about macular degeneration genetic risk, [...] This is an exciting and overall potentially very important study and is of interest to readers of Nat Comm.

Reply: We thank the reviewer for their supportive appraisal of this exciting work.

[...] Preincubating the antibody with FHL-1 does not block labeling with the FHR-4 antibody, but how about preincubation with full length FH? Do the authors know where the epitope for Clone 150 resides? What if it corresponds to a domain of FH that's absent from FHL? [...] there is concern that even though the MAbs may have much lower affinity for FH compared to FHR-4, the much more abundant FH protein may be responsible for the signal detected on IHC.

Reply: We have performed many experiments to confirm the specificity of the FHR-4 antibodies, and consistently find no cross-reactivity with FH. We do not know the precise epitope recognised by clone 150 but plan to map all the mAbs in the future. Although in the data shown we did not pre-incubate the anti-FHR-4 antibody with full length FH, we did include FH in the western blotting analysis in Supplementary Figure 10B which demonstrated no cross-reactivity with either clone used (in ELISA or IHC). Also, the staining pattern of FHR-4 in the intercapillary septa of the choriocapillaris is not consistent with FH staining seen previously (Clark *et al.* (2014)). To fully address this concern, we now include in the revised manuscript additional ELISA data demonstrating that binding of clone 150 to immobilised FHR-4 protein can be successfully competed with increasing concentrations of fluid-phase FHR-4 protein - but fluid-phase FH, even at a 100-fold molar excess to the competing dose of FHR-4, has no competitive effect, conclusively demonstrating that there is no cross-reactivity of the mAb with FH (please see new Supplementary Figure 11).

This should ideally be demonstrated using protein biochemistry, especially precipitation of FH and FHR-4 from the RPE/Bruch's/choroid and MS of the pulled down fractions and minimally demonstration of FHR-4 and FH and FHL-1 levels in choroid protein extracts.

Reply: As discussed above, it is not possible to isolate the extracellular matrix of the choroid and Bruch's membrane from the vascular compartment (and the choroid is a highly vascular tissue) to reliably measure levels of complement proteins. Furthermore, the use of mass spectrometry to measure the levels of these homologous proteins is challenging, particularly, for example, to distinguish FH and FHL-1.

A few smaller issues to consider are listed below:

1. The conclusion in the abstract that FHR-4 is the key molecular player in AMD is overstated given the data.

Reply: This is addressed in the new (as per journal's guidelines 150-word) abstract, edited title and modified final paragraph of the discussion.

2. Details on the patient and control cohort would be helpful. While the authors state that the statistics are adjusted for age, how much adjustment was necessary? If there was not overlap between the cases and controls, it may not be possible to correct for that with linear regression. For that matter, was FHR-4 level related to age at all in the dataset?

Reply: We thank the reviewer for this comment; although complement activity has been observed to change with age (Gaya da Costa et al., 2018, Lorés-Motta et al., 2018), we had not evaluated this in the case of FHR-4 levels. In our dataset, FHR-4 levels are not correlated with age in any of the two cohorts (Cambridge: Spearman $\rho_{\text{AMD}\&\text{controls}}=0.060$, $P\text{-value}_{\text{AMD}\&\text{controls}}=0.173$, Spearman $\rho_{\text{controls}}=0.072$, $P\text{-value}_{\text{controls}}=0.297$; EUGENDA: Spearman $\rho_{\text{AMD}\&\text{controls}}=0.043$, $P\text{-value}_{\text{AMD}\&\text{controls}}=0.338$, Spearman $\rho_{\text{controls}}=-0.049$, $P\text{-value}_{\text{controls}}=0.393$), so indeed adjustment might have not been necessary. However, there is an overlap between the ages for both cohorts (see figure below) and, therefore, the raw estimates with no adjustment for age were comparable: Cambridge: $\text{Beta}_{\text{unadjusted}}=0.18$, $\text{SE}_{\text{unadjusted}}=0.07$, $\text{Beta}_{\text{adjusted}}=0.17$, $\text{SE}_{\text{adjusted}}=0.07$; EUGENDA: $\text{Beta}_{\text{unadjusted}}=0.19$, $\text{SE}_{\text{unadjusted}}=0.05$, $\text{Beta}_{\text{adjusted}}=0.24$, $\text{SE}_{\text{adjusted}}=0.06$). Unadjusted and adjusted Beta and SE values have been added to Table 1 for both association analyses of FHR-4 and FH levels with AMD.

3. A cartoon of the difference between *fH*, *FHL1* and *FHR4* would be helpful to educate the reader and especially with the interpretation of the antibody and blocking experiments.

Reply: We have now referred the reader to Figure 3 of one of our previously published papers [Clark, S.J. and Bishop, P.N. (2015) Role of factor H and related proteins in regulating complement activation in the macula. *J. Clin. Med.* **4**, 18-31] for an explanatory diagram of *CFH* and *CFHR* genes and the structures of FH, FHL-1 and FHR proteins (lines 75-76).

4. The authors say a few places that they used “phenotyped” donor tissue—does the phenotyping come into play otherwise in the manuscript?

Reply: The reviewer makes a very good point. Actually, no - the phenotyped status of the tissue doesn't come into play in the manuscript, other than demonstrate that the source of material (the Manchester Eye Tissue Repository, or METR) provides well characterised samples. Indeed the two instances of ‘phenotyped’ have now been dropped in the required new 150-word abstract and new methods section in the main manuscript.

5. The donor tissue is up to 48 hours after death. Can the authors convince the reader that this is an acceptable interval for IHC?

Reply: We are not aware of any evidence to suggest that these post-mortem times will affect results of IHC. Indeed, we have published several other papers showing that we can reliably and reproducibly localise complement and other proteins in ocular tissue with these post mortem times. Examples include (1) Clark SJ, Schmidt CQ, White AM, Hakobyan S, Morgan BP, Bishop PN. Identification of factor H-like protein 1 as the predominant complement regulator in Bruch's membrane: implications for age-related macular degeneration. *J Immunol.* 2014 Nov 15;193(10):4962-70. doi: 10.4049/jimmunol.1401613. (2) Clark SJ, Ridge LA, Herbert AP, Hakobyan S, Mulloy B, Lennon R, Würzner R, Morgan BP, Uhrin D, Bishop PN, Day AJ. Tissue-specific host recognition by complement factor H is mediated by differential activities of its glycosaminoglycan-binding regions. *J Immunol.* 2013 Mar 1;190(5):2049-57. doi: 10.4049/jimmunol.1201751. (3) Keenan TD, Clark SJ, Unwin RD, Ridge LA, Day AJ, Bishop PN. Mapping the differential distribution of proteoglycan core proteins in the adult human retina, choroid, and sclera. *Invest Ophthalmol Vis Sci.* 2012 Nov 7;53(12):7528-38. doi: 10.1167/iovs.12-10797. (4) Clark SJ, Perveen R, Hakobyan S, Morgan BP, Sim RB, Bishop PN, Day AJ. Impaired binding of the age-related macular degeneration-associated complement factor H 402H allotype to Bruch's membrane in human retina. *J Biol Chem.* 2010 Sep 24;285(39):30192-202. doi: 10.1074/jbc.M110.103986.

6. *The quantitative aspects of the study are a little confusing. The authors note that a 2.5x molar excess of FHR-4 over FHL-1 caused 50% reduction in C3b cleavage. Is this a ratio that is likely to ever exist in nature in Bruch's/choroid? Again the biochemical demonstration of the abundance of these factors would be valuable*

Reply: The *in vitro* biochemical experiments put into context what increasing concentrations of FHR-4 would mean to the capacity of FHL-1 to confer complement regulation on C3b. Unfortunately, it is not currently possible to measure the exact amount of each protein in the choriocapillaris of the human eye (as mentioned in reply to your previous question).

7. *Figure 3B- the authors are encouraged to please look at this figure carefully—the legends seem miniscule to this reader (although the data are great)—please remake this*

Reply: We apologise for the issue. The whole figure 3 has now been split into three different figures (new Figure 3, Figure 5 and Figure 6) using the original panels and corresponding new figure legends have been provided.

8. *It's a little concerning that advanced AMD is treated like a single entity. While the authors stratified based on genotype (and those data are very compelling) there is doubt as to whether CNV and GA should be grouped together. Can the authors split these and show if the signal is coming from one or both end stages?*

Reply: We appreciate the concern of the reviewer as, phenotypically, the two end-stages of AMD choroidal neovascularization (CNV) and geographic atrophy (GA) are distinct. However, the largest genome-wide association study on advanced AMD (Fritsche et al., 2016, IAMDC study) has shown that, genetically, these two phenotypes are not as distinct (“heritability estimates for choroidal neovascularization ($h^2 = 44.3\%$, $CI = 42.2-46.5\%$) and geographic atrophy ($h^2 = 52.3\%$, $CI = 47.2-57.4\%$) were similar; bivariate analyses showed a genetic correlation of 0.85 ($CI = 0.78-0.92$) between these disease subtypes”); all lead variants at the established 34 AMD-associated loci were associated with both GA and CNV in the IAMDC study, with variant rs42450006 upstream of *MMP9* being the only one

exclusively associated with CNV (frequency in controls = 14.1%, $P = 8.4 \times 10^{-17}$, OR = 0.78) but not with GA ($P = 0.39$, OR = 1.04; $P_{\text{difference}} = 4.1 \times 10^{-10}$). No differences in the genetic risk between CNV and GA have been shown for the 8 established *CFH* locus variants.

Nevertheless, we agree that a stratified analysis can remain of interest, but it has to be noted that our study design has reduced power for such analysis, especially for the ‘GA only’ group (n=62 in the Cambridge cohort and n=10 in the EUGENDA cohort). The results from the phenotype-based analysis are summarised in the table below and the corresponding text has been added to lines 113-120.

We hope that the novelty and importance of our findings will trigger other research groups worldwide to measure FHR-4 levels in additional samples and allow larger studies in the near future.

Association of FHR-4 levels with AMD			
	CNV only	GA only	All AMD
	Beta, SE, P	Beta, SE, P	Beta, SE, P
Cambridge			
Unadjusted	0.15, 0.08, 0.068	0.20, 0.12, 0.099	0.18, 0.07, 0.016
Adjusted*	0.12, 0.08, 0.116	0.20, 0.11, 0.082	0.17, 0.07, 0.018
EUGENDA			
Unadjusted	0.18, 0.05, 0.001	0.45, 0.17, 0.008	0.19, 0.05, 1.7×10^{-4}
Adjusted*	0.23, 0.06, 2.5×10^{-4}	0.49, 0.17, 0.005	0.24, 0.06, 8.4×10^{-5}
Association of FHR-4 levels with AMD			
	CNV only	GA only	All AMD
	Beta (95% CI), P	Beta, 95% CI, P	Beta, 95% CI, P
Meta-Analysis			
Unadjusted	0.17 (0.09-0.26), 9.3×10^{-5}	0.28 (0.09 – 0.47), 0.0037	0.19 (0.11 – 0.27), 7.1×10^{-6}
Adjusted*	0.19 (0.09 – 0.28), 1.0×10^{-4}	0.29 (0.10 – 0.48), 0.0024	0.21 (0.12 – 0.30), 4.8×10^{-6}

*Adjusted for sex, age, batch effects and first two genetic principal components

9. The authors note that the circulating levels of FHR-4 are low but that the ratios of FHR-4 to FH/FHL-1 are probably different in serum than they are in the ECM of Bruch’s. This is probably true--but it’s not obvious that this ratio is tipped in a way that favors more FHR-4 on the ECM, especially with the high degree of local synthesis of FH in the RPE and the choroid. The authors are encouraged to determine this ratio with biochemistry.

Reply: As discussed above it is not possible to measure local levels, but we infer that raised systemic levels of FHR-4 will result in increased local levels in the extracellular matrix of the choriocapillaris. A major contributing factor is the selective permeability of Bruch’s membrane, where we already know that smaller proteins can diffuse through easier, thus excluding FH and tipping the ratio towards more FHR-4.

10. One of the interesting implications from this work is that local synthesis in the eye of complement inhibitors is less important than the circulating regulators (i.e., FHR-4). This reviewer feels the jury is certainly out on that question, but wonders if the authors can address the recent studies by Khandhadia and colleagues that suggest liver transplantation genotype is less impactful on AMD risk compared to host somatic genotype.

Reply: We agree that the jury is still out on this question. Our data provides evidence that systemically derived FHR-4 is important. However, we do know that other complement proteins, including FH, FHL-1, C3 and FI are synthesised locally, so for these proteins it is entirely possible that both locally synthesis and systemically produced proteins are important. We should emphasise here, however, that *CFHR4* gene transcription has only ever been observed in the liver so in this case FHR-4 provides a systemic contribution to the disease.

Reviewers' comments:

Reviewer #1 (Remarks to the Author):

The authors have been very responsive to the previous round of review. I have no further substantial comments.

Minor points for consideration for potential reader interest:

1. On line 159, the authors observed a single participant carrying the CFH 1210C allele in an AMD case from Cambridge, but excluded this individual from analysis. Could the authors consider showing the serum FH and FHR-4 levels for this patient, in light of the presence of a special, rather highly penetrant mutation carried by this patient? Will it be different from the rest of the participants? Or will there be no difference? Either way, the data could be of interest, even though presented in an exploratory manner (N=1). There should be no need for statistical treatment in this case.

2. Is it possible to perform a mediation analysis between:

- a) the genome-wide significant SNPs at the broad CFH locus,
- b) FHR-4 levels
- c) FH-levels?

CC Khor

Reviewer #3 (Remarks to the Author):

In the revised manuscript, the authors were responsive to the previous reviews and have provided important additional data.

The authors are congratulated on this excellent study. This manuscript is of very high interest to the field, is translationally very significant, uses large data sets with compelling demonstration of replicability, and would be of high interest to readers of a broad, high impact journal.

One issue that could still be addressed relates to the FHR4/FH antibody question. This reviewer agrees that the authors have largely done due diligence to the (enormously important) antibody specificity question. The addition of the ELISA figure (supplemental 11) is convincing, although there is still the unfortunate but well appreciated observation that many antibodies behave differently in different platforms (Western blot vs ELISA vs IHC). It is not implausible that full length FH has C-terminal epitopes recognized by the anti-FHR4 antibodies on tissue sections (especially if FH is hundreds of times more abundant in tissue), and this possibility could be more fully addressed.

The authors argue that this is not possible because FHR4 (current paper) and FH (from their previous papers) have different binding patterns in Bruch's/the choroid. The concern is that those experiments, performed at a different time and perhaps on different samples may not be directly comparable.

In order to fully clinch this, the authors are encouraged to show dual labeling of the FHR4 and FH antibodies on the same sections. While only semi-quantitative, this could also give an indirect sense of the relative abundance of these molecules.

This is a question only pertaining to the IHC, as the ELISA and Western data showing a compelling association of circulating FHR4 levels (which comprise the most critical observations in the paper) are rock solid.

Minor-

In supplemental Figure 10, please specify the meaning of the "FH" lane. This figure might also be clarified by showing the relative position on the blot of FH and FHL1 (on separate lanes of the same experiment) and the pattern if primary antibody is omitted (is this the FH lane?).

Reviewer #1

The authors have been very responsive to the previous round of review. I have no further substantial comments.

Reply:

We thank the reviewer for their time spent on reviewing our manuscript and their comments helping us consolidating even further our findings.

Minor points for consideration for potential reader interest:

1. On line 159, the authors observed a single participant carrying the CFH 1210C allele in an AMD case from Cambridge, but excluded this individual from analysis. Could the authors consider showing the serum FH and FHR-4 levels for this patient, in light of the presence of a special, rather highly penetrant mutation carried by this patient? Will it be different from the rest of the participants? Or will there be no difference? Either way, the data could be of interest, even though presented in an exploratory manner (N=1). There should be no need for statistical treatment in this case.

Reply:

We thank the reviewer for triggering this further analysis of our data. We have now added to the main text the FHR-4 and FH levels corresponding to the single Cambridge AMD-case R1210C carrier in our dataset, which are equal to 5.7 and 296.4, respectively (lines 161-162). Interestingly, both values seem indeed different from the rest of the Cambridge case participants, i.e., below the case FHR-4 95% CI (i.e., 6.0-7.2) and within the control FHR-4 95% CI (i.e., 4.9-6.2), and below the FH 95% CIs for both the case set (340.2-357.2) and the control set (i.e., 338.9-359.4), respectively. Nevertheless, as the reviewer suggested, we are not engaging in any statistical analysis (given N=1), but we note that this finding may well generate further research hypotheses in the readers.

2. Is it possible to perform a mediation analysis between:

- a) the genome-wide significant SNPs at the broad CFH locus,*
- b) FHR-4 levels*
- c) FH-levels?*

Reply:

We agree with the reviewer that our findings generate interest for new analyses, and we see that as further strength of our study. We believe that the mediation analysis suggested can be best pursued by conducting a bold, large-scale, 2-sample Mendelian Randomization study, both on the causal role of FHR-4 levels in AMD, as a univariate analysis, as well as a multivariate analysis of FHR-4 and FH levels together, and any other relevant biomarkers and/or measurements of the complementome. Indeed, we have been already proactive in planning further research grant applications and setting up collaborations with other members from the International AMD Genomics Consortium, as well as other research groups in relevant related fields, to take forward our results and perform such analyses in future larger studies.

Finally, we expect our work to be game-changing as it unveils FHR-4 as a novel, major player beyond FH in AMD pathogenesis. We believe that the publication of our findings will trigger the measurement of FHR-4 levels in large AMD cohorts available

worldwide and/or other large population-based cohorts, and that will in turn facilitate any more complex association analyses.

Reviewer #3

The authors are congratulated on this excellent study. This manuscript is of very high interest to the field, is translationally very significant, uses large data sets with compelling demonstration of replicability, and would be of high interest to readers of a broad, high impact journal.

Reply:

We thank the reviewer for their recognition of our work.

One issue that could still be addressed relates to the FHR4/FH antibody question. This reviewer agrees that the authors have largely done due diligence to the (enormously important) antibody specificity question. The addition of the ELISA figure (supplemental 11) is convincing, although there is still the unfortunate but well appreciated observation that many antibodies behave differently in different platforms (Western blot vs ELISA vs IHC). It is not implausible that full length FH has C-terminal epitopes recognized by the anti-FHR4 antibodies on tissue sections (especially if FH is hundreds of times more abundant in tissue), and this possibility could be more fully addressed.

Reply:

Respectfully, we feel that we have already conducted more than necessary levels of due diligence to characterise the anti-FHR-4 antibody, and its lack of cross-reactivity with FH. We already show that clone 150 (the Ab used in the IHC studies) does not recognise either pure FH, or FH in the context of whole human serum (where only the A and B variants of the FHR-4 protein are detected).

Furthermore, the reviewer's concern has already been addressed in the additional competition ELISA we performed (supplementary figure 11) where, by the reviewer's own admission, the evidence that excessive amounts of FH cannot out-compete FHR-4 binding to the IHC Ab, thus demonstrating the specificity of the Ab is 'rock-solid'. FH associated with ECM in tissue sections will not suddenly make it able to interfere with anti-FHR-4 binding to FHR-4.

In order to fully clinch this, the authors are encouraged to show dual labeling of the FHR4 and FH antibodies on the same sections. While only semi-quantitative, this could also give an indirect sense of the relative abundance of these molecules.

Reply:

We argue that this suggested experiment won't actually add anything to the extensive characterisation of the clone 150 antibody we have already undertaken. The suggested semi-quantitative analysis is unlikely to add anything as we already know that FH/FHL-1 and FHR-4 occupy very similar space in the ECM of the choriocapillaris, therefore co-localisation would be expected. Even if multiple z-plane confocal microscopy images were to be obtained we very much doubt that it would allow sufficient resolution in the tissue to add useful information on antibody binding specificity. Therefore, we think that this experiment will not be useful and will not add anything to our already extensive characterisation of the clone 150 monoclonal antibody.

Minor-

In supplemental Figure 10, please specify the meaning of the “FH” lane. This figure might also be clarified by showing the relative position on the blot of FH and FHL1 (on separate lanes of the same experiment) and the pattern if primary antibody is omitted (is this the FH lane?).

Reply:

In this figure the lane labelled FH has purified FH loaded and is there to demonstrate that none of the anti-FHR-4 clones used in this study cross-reacted with FH. We have adjusted the legend to make this clearer.